# Impact of visual callosal pathway is dependent upon ipsilateral thalamus

Vishnudev Ramachandra[1], Verena Pawlak[1], Damian J. Wallace[1] & Jason N. D. Kerr[1✉]

The visual callosal pathway, which reciprocally connects the primary visual cortices, is thought to play a pivotal role in cortical binocular processing. In rodents, the functional role of this pathway is largely unknown. Here, we measure visual cortex spiking responses to visual stimulation using population calcium imaging and functionally isolate visual pathways originating from either eye. We show that callosal pathway inhibition significantly reduced spiking responses in binocular and monocular neurons and abolished spiking in many cases. However, once isolated by blocking ipsilateral visual thalamus, callosal pathway activation alone is not sufficient to drive evoked cortical responses. We show that the visual callosal pathway relays activity from both eyes via both ipsilateral and contralateral visual pathways to monocular and binocular neurons and works in concert with ipsilateral thalamus in generating stimulus evoked activity. This shows a much greater role of the rodent callosal pathway in cortical processing than previously thought.

[1] Department of Behavior and Brain Organization, Research Center caesar, 53175 Bonn, Germany. ✉email: jason.kerr@caesar.de

At the earliest stages of cortical visual processing visually responsive neurons can be divided into those only responsive to one eye (monocular neurons) and those responsive to both eyes (binocular neurons). For binocular neurons, the most direct anatomical pathway giving rise to responses is through the ipsilateral lateral geniculate nucleus (LGN)[1,2]. Cortical responses to contralateral eye stimulation arise primarily from its crossed projection to the ipsilateral LGN and the subsequent projection from the LGN to visual cortex, while cortical responses to the ipsilateral eye mainly arise from its uncrossed projection to the ipsilateral LGN (see Fig. 1 for schematic). An alternative anatomical pathway giving rise to responses from the ipsilateral eye results from its tri-synaptic projection involving the contralateral visual cortex and projection through the corpus callosum (ipsilateral eye–contralateral LGN–contralateral visual cortex–ipsilateral visual cortex via the callosal projection)[3–5]. Such interhemispherically projecting neurons can be found in many mammalian species[4,6–10] including rodents[6,11,12], where a subpopulation of neurons project their axons to the contralateral visual cortex. These callosal pathway projecting neurons originate from, and in turn target, the visual cortex regions where receptive fields are activated by stimuli presented along the vertical meridian[3,4,8,13], in front of the animal. In cats and primates, where the visual callosal pathway has been extensively investigated[3,4,10], the callosal pathway has been proposed to facilitate stereovision[3,4] through the binocular pathway. However, differences in rodent visual system, compared to cat and primate, at the level of the projections from the retinae[14,15] and the central visual centers[16,17], hinder clear comparisons of callosal function. One suggestion for the function of the callosal projection is that it facilitates lateral interactions between neurons representing adjacent regions in the visual field across the location where the visual field is divided between the hemispheres[8,18,19]. This has been proposed for primates and cats where the visual field is divided along the vertical meridian with the left and right halves of the visual field (hemifields) being represented in the right and left hemispheres, respectively. However, in many other animals including rodents[8,16,20,21], there is considerably more overlap in the representation of the visual field around the vertical meridian in the two cortical hemispheres, and lateral connectivity can in principle be achieved without the callosal projection. Single-cell recordings in rodents have suggested that the callosal pathway contributes ipsilateral eye-derived visual inputs[13,22,23] for binocular processing, while other studies have suggested that the callosal pathway has little if any influence on ipsilateral eye activation of visual cortex neurons[11,24]. While numerous studies have investigated the contribution of the callosal pathway to stimulus activation of binocular neurons[11,13,22,23,25], it is not known what the relative contribution of the isolated visual pathway is in generating responses across neuronal populations, including the activation of monocular neurons.

Here, we used optogenetic and pharmacological manipulation combined with multiphoton population imaging and retrograde anatomical tracing, to quantify the role of the visual callosal pathway on spiking responses to visual stimulation of neurons located in the binocular–cortex. We show that visual cortex neurons projecting their axons through the callosal pathway can modulate spiking responses in subpopulations of both monocular and binocular neurons in the opposite cortex. At the individual cell level, there were a wide range of possible outcomes of blocking this pathway during visual stimulation. At one extreme subpopulations of monocular and binocular neurons completely lost their ability to generate spiking responses to visual stimuli upon callosal pathway inactivation, at the other extreme callosal pathway inactivation had no effect on a subpopulation of monocular and binocular neurons spiking response. We also

show that while blocking this pathway significantly reduced spiking across neuronal populations, the callosal projection alone was not capable of driving suprathreshold activity in V1 neurons, but required simultaneous input from the ipsilateral LGN. Together this shows that the callosal pathway plays an essential role in shaping cortical spiking responses to visual stimuli presented to either eye, but only in conjunction with ipsilateral LGN activation.

## Results

**Locating the visual callosal pathway and measuring its role in visual cortex activity.** To locate the visual callosal pathway (vc pathway), we injected retrograde tracers (alexa-conjugated cholera toxin beta subunit, CTB) into multiple sites in the primary visual area (Fig. 1a). Injections into the binocular area (V1b) resulted in retrograde labeling of contralateral hemisphere neuronal somata along a narrow band at the border of primary (V1) and secondary (V2) visual areas (FWHM in medial–lateral dimension: $0.72 \pm 0.17$ mm, mean $\pm$ SD, $N = 3$, Fig. 1a, b). Separate injections targeted to surrounding monocular regions did not result in labeling of neurons in contralateral cortex (Fig. 1a), supporting previous studies showing callosal projections are limited to lateral V1 (refs. [6,12]). In this labeled region we used multiphoton imaging of $Ca^{2+}$ transients to quantify L2/3 neurons' suprathreshold responses to monocular and binocular visual stimulations (Fig. 1c, d for experimental setup and terminology, Fig. 1e, f). Labeled neurons could be statistically classified as ipsilaterally or contralaterally responsive only or as binocularly responsive (Fig. 1g) by first converting the recorded $Ca^{2+}$-responses to spiking activity[26] (Fig. 1f) and comparing stimulus-locked responses to baseline spontaneous activity (see "Methods" section for statistical criteria). All recorded fields of view contained both visually responsive and non-responsive neurons (147/279 responsive, 132/279 non-responsive, FOV = 22, $N = 17$ animals), with responsive neurons comprising monocular ipsilaterally and contralaterally responsive neurons (Fig. 1h, i) and binocularly responsive neurons (19% ipsilateral, 42.9% contralateral, and 38.1% binocular). Based on intrinsic optical signal imaging response (see "Methods" section and Supplementary Fig. 1), all fields of view were overlaid showing that both monocular and binocular neurons were interspersed and showed no obvious spatial clustering (two-tailed Fisher's exact test $P = 0.5647$, Supplementary Fig. 2). We next quantified functional properties of projecting and non-projecting neurons.

**VC pathway mirrors stimulus-evoked activity contralaterally.** Responses were recorded from neuronal populations after retrogradely labeling VCPNs (Fig. 2a, $N = 3$ animals, $N = 5$ FOV, $N = 100$ neurons). The first co-labeled neurons were located ~140 μm below the pia (Fig. 2b, Supplementary Fig. 3). Within layer 2/3, the VCPNs were the minority of the neuronal population (34.76 $\pm$ 5.67%, mean $\pm$ SD, $N = 362$ neurons, Fig. 2b), and often VCPNs and non-VCPNs were direct neighbors (Fig. 2a) without showing specific spatial clustering (pairwise distance distribution: VCPNs vs. randomly chosen neurons of the same number as that of VCPNs, Kolmogorov–Smirnov test $P = 0.176$; non-VCPNs vs. randomly chosen neurons of the same number as that of non-VCPNs, Kolmogorov–Smirnov test $P = 0.996$). The mean spiking rate of the VCPN subpopulation evoked by drifting grating stimuli was not significantly different to that of non-VCPN subpopulation (VCPN vs. non-VCPN, $0.32 \pm 0.22$ vs. $0.4 \pm 0.29$ Hz, mean $\pm$ SD, Mann–Whitney $U$ test $P = 0.437$, $N = 28$ (VCPN) and $N = 33$ (non-VCPN), gratings presented in the visual space in front of the nose, see "Methods" section for details). Mean spontaneous firing rates were also not different

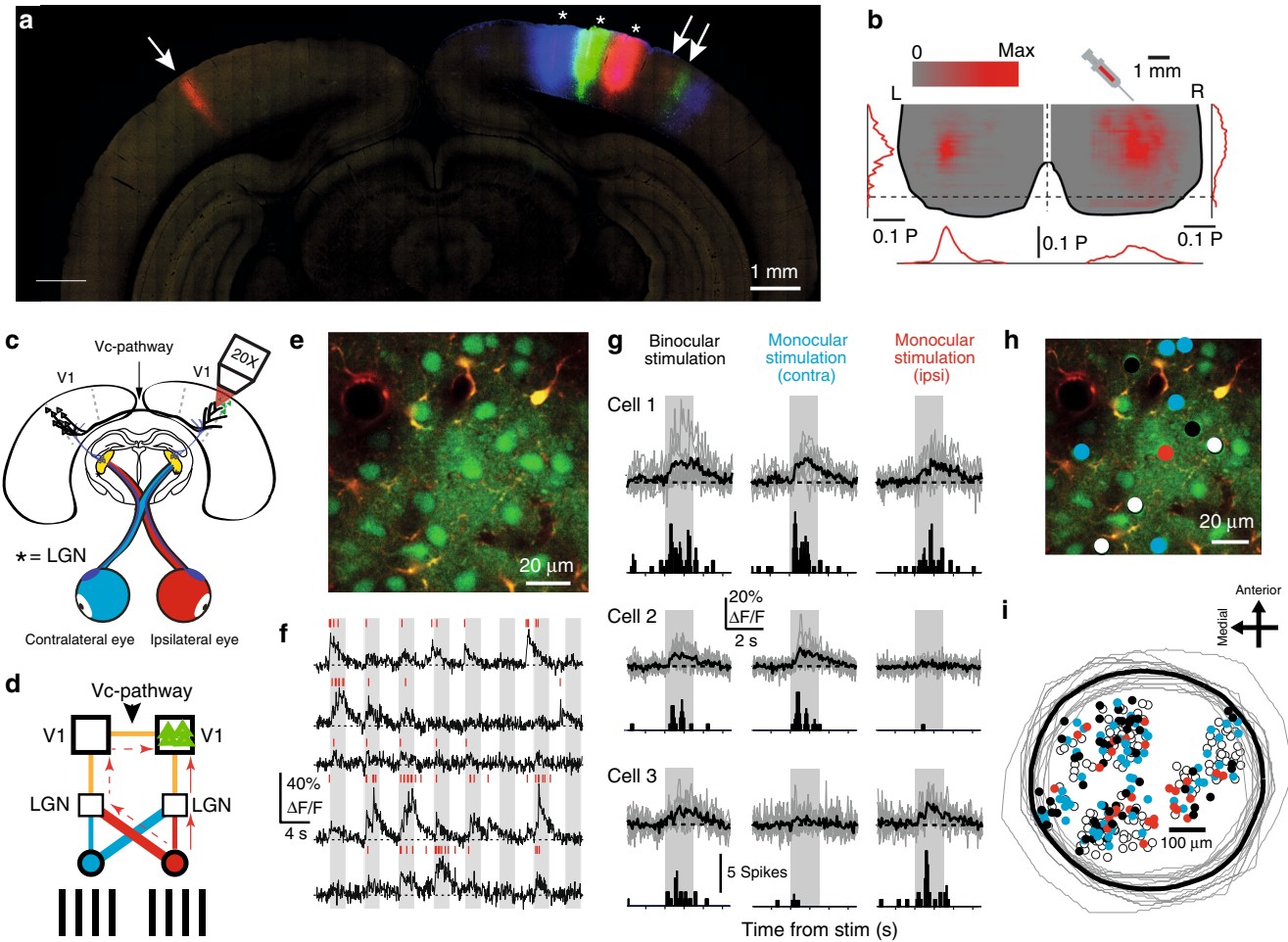

**Fig. 1 Localizing the visual callosal pathway and experimental setup. a** Coronal section of rat brain showing fluorescence from CTB-Alexa 594 (red), CTB-Alexa 488 (green), and CTB-Alexa 647 (blue) dyes injected into rat visual cortex (asterisks) at 4.1, 3.0, and 2.0 mm lateral to midline. Note that injections resulted in retrograde labeling of contralateral neurons (red, white single arrow), and retrograde labeling of neurons in V2 (red, green, blue, white double arrow). **b** Reconstruction of left and right occipital poles showing average regions of retrograde labeling of CTB-Alexa 594 showing coronal and sagittal pixel intensity distributions ($N = 3$ animals). **c** Schematic of experimental design. **d** Visual stimulation convention showing major visual pathways from eyes (circles) to dorsal LGN (small squares) to cortical V1 (large squares) and visual callosal pathway (vc pathway). Red arrows indicate the two potential anatomical pathways to ipsilateral V1 from the ipsilateral eye, direct via ipsi-LGN (solid) and via the callosal pathway (dashed). **e** Overview example 2-photon image showing functionally labeled neurons (green) and astrocytes (red). **f** Both spontaneous (between gray boxes) and evoked (gray boxes) $Ca^{2+}$-transients from five neurons in response to moving gratings at different angles (angles not shown). Inferred spiking activity within an imaging frame (red marks for single APs) and the underlying $Ca^{2+}$-transients shown. **g** Activity traces from three representative neurons during binocular stimulation (left), contralateral stimulation (middle) and ipsilateral stimulation (right). Calcium responses to single stimuli are overlaid (gray) with average (black) and inferred-spike PSTH (lower to each trace). Binocular neuron (cell 1), contralateral stimuli responsive monocular neuron (cell 2) and ipsilateral stimuli responsive monocular neuron (cell 3). **h** Overview example 2-photon image (same as **e**) labeled with response types within a field of view, non-responders (white), contralateral stimuli responsive monocular (turquoise), ipsilateral stimuli responsive monocular (red) and binocular (black). **i** Neuronal response type maps overlaid using intrinsic image responses ($N = 17$ animals, 22 non-overlapping imaging areas, contains example shown in **h**). Each intrinsic imaging area (gray) was centered and associated imaged population overlaid, average intrinsic border (black).

(VCPN vs. non-VCPN, $0.17 \pm 0.14$ vs. $0.16 \pm 0.16$ Hz, mean ± SD, Mann–Whitney $U$ test $P = 0.599$, Fig. 2c). While the maximum response to the preferred orientation was not significantly different (mean of contralateral and ipsilateral eye preferred orientation response for binocular neurons, preferred orientation response for monocular neurons, VCPN vs. non-VCPN, $1.0 \pm 0.46$ Hz vs. $1.41 \pm 1.08$ Hz, mean ± SD, Mann–Whitney $U$ test $P = 0.342$, Fig. 2c), a significantly higher fraction of VCPNs were responsive to drifting grating stimuli (VCPN vs. non-VCPN, $77.46 \pm 3.37\%$ vs. $55.91 \pm 7.56\%$, mean ± SEM, two-tailed Fisher's exact test $P = 0.0329$). The distributions of preferred orientations between the two groups were the same, indicating that this area

transmits an unbiased representation of the orientation-relevant stimulus responses (contralateral eye response: VCPN vs. non-VCPN, $112.0° \pm 34.19°$ vs. $137.98° \pm 45.23°$, circular mean ± SD, Kuiper test $P > 0.1$, $N = 14$ VCPN and 18 non-VCPN; ipsilateral eye response: VCPN vs. non-VCPN, $113.22° \pm 43.65°$ vs. $115.81° \pm 45.78°$, circular mean ± SD, Kuiper test $P > 0.1$, $N = 14$ VCPN and 20 non-VCPN, Fig. 2d). Ocular dominance distributions were also not significantly different (contralateral bias index; VCPN vs. non-VCPN, $0.582 \pm 0.186$ vs. $0.577 \pm 0.177$, median ± SD, Kolmogorov–Smirnov test $P = 0.734$, Fig. 2e). The VCPNs and non-VCPNs identified by this retrograde tracing method were functionally indistinguishable from each other, suggesting

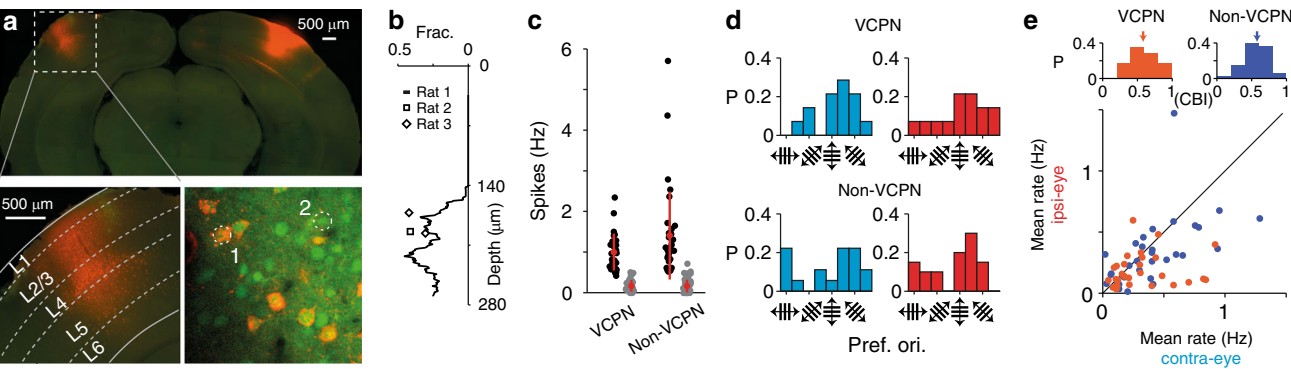

**Fig. 2 Functional responses of projecting and non-projecting neurons are indistinguishable. a** Coronal rat brain section showing the CTB-Alexa 594 injection site and retrogradely labeled neurons in contralateral V1 (upper) expanded view of retrogradely labeled projection neurons location across cortical layers (lower left, from upper image). Lower right: Overview of neurons labeled with OGB-1 (green, e.g. neuron 2) with callosal projecting neurons (VCPNs) co-labeled with CTB (red, e.g. neuron 1). **b** Fraction of VCPNs as a function of depth from the pia ($N = 3$ animals). **c** Pooled data of spontaneous (gray) and maximum stimulus response (black, average max. response to preferred orientation for VCPNs ($N = 28$ neurons) and non-VCPNs ($N = 33$ neurons)) and mean ± SD (red) for each group. **d** Distribution of preferred orientations for orientation selective neurons from both VCPN and non-VCPN groups (ipsi-eye stimulation, red, contra, turquoise). **e** Scatter plot comparing mean stimulus response of ipsi- to the mean response from the contralateral eye stimulation (lower) for VCPNs (orange markers) and non-VCPNs (blue markers) with the distribution of the contralateral bias index shown in upper panels (see "Methods" section for details).

that, at least for the stimulus types tested here, the vc pathway mirrors stimulus-evoked information to the opposite hemisphere. To test the impact of vc pathway, we next optogenetically modulated this pathway during visual stimulation.

**Vc pathway modulates monocular and binocular spiking responses**. We used an adeno-associated virus (AAV) to express the optogenetic activity inhibitor eArchT 3.0 (ref. [27]) and a yellow-fluorescent protein (YFP) tag in neurons in the region containing the VCPNs (Fig. 3a, Supplementary Fig. 4, $N = 29$ animals). This simultaneously established the location of the termination of callosal projections to contralateral hemisphere (Fig. 3a). Labeled axonal arbors formed dense arborizations localized around the V1–V2 border region spanning all cortical layers (Fig. 3a). We next identified cortical representation of the binocular visual space directly in front of the animal using intrinsic optical signal imaging, and made a small injection of Alexa-conjugated CTB into layer 2/3 in the center of the identified region (small red spot in Fig. 3a, see "Methods" for full details). This showed that the V1–V2 border region receiving the afferent callosal projection overlaps with the representation of the frontal binocular visual field. Neurons expressing eArchT 3.0 were located across all cortical layers, and juxtasomal electrical recordings confirmed that activation of eArchT rapidly and reversibly silenced both spontaneous and stimulus-evoked firing in neurons in all cortical layers (Fig. 3b, c, Supplementary Fig. 4, 22 recordings, $N = 10$ animals). We next quantified the extent to which inhibition of the vc pathway modulated suprathreshold activity to visual stimuli across neuronal populations (Fig. 3d).

We compared the response probabilities for visually responsive neurons to monocular and binocular visual stimulations, with and without vc pathway inactivation (Fig. 3e, total of 41 neurons, 14 binocular, 9 ipsilateral, 18 contralateral responsive neurons, $N = 6$ populations, $N = 5$ animals). Average response probability of visually responsive neurons across all stimulation conditions was 0.346 ± 0.19, mean ± SD (binocular stimulation 0.406 ± 0.206, ipsilateral stimulation 0.255 ± 0.177, contralateral stimulation 0.377 ± 0.229, $N = 41$ neurons), and 0.605 ± 0.204 for the maximally tuned response angle (binocular stimulation 0.773 ± 0.213, ipsilateral stimulation 0.562 ± 0.264, contralateral stimulation 0.746 ± 0.246, $N = 41$ neurons). Inactivating the vc pathway

significantly reduced both the neurons' average response probabilities when taking the response to all presented angles (0.277 ± 0.166, mean ± SD, paired Wilcoxon signed-rank test $P < 0.0001$; binocular stimulation 0.34 ± 0.191, ipsilateral stimulation 0.184 ± 0.147, contralateral stimulation 0.308 ± 0.209), as well as the average response probabilities for the neurons' maximally tuned response angle (0.506 ± 0.21, mean ± SD, paired Wilcoxon signed-rank test $P < 0.0001$; binocular stimulation 0.688 ± 0.25, ipsilateral stimulation 0.466 ± 0.269, contralateral stimulation 0.614 ± 0.286, Supplementary Fig. 5a). The vc pathway has been implicated in the generation of correlated firing responses between the cortices[28]. We therefore next investigated the possibility that the vc pathway is involved in generating network synchrony by influencing correlated activity. Probability distributions of pairwise correlations were not significantly changed by vc pathway inactivation (Supplementary Fig. 5b, $P = 0.221$), but as pairwise correlation does not capture a complete view of correlated population-wide spiking events[26], we next quantified how the visually active neurons represented binocular and monocular stimulations as a population.

We quantified, for visually responsive neurons, how many spikes were elicited from each single stimulus presentation for both monocular and binocular stimuli as a measure of the strength of each pathway in generating population-wide correlated spiking events[26] (Fig. 3f). Monocular contralateral and binocular stimuli elicited similar probability distributions, while ipsilateral stimulation generated significantly less spiking and more trials with no spikes, which was also significantly more than observed during spontaneous activity (binocular vs. contralateral, Kolmogorov–Smirnov test $P = 0.299$; binocular vs. ipsilateral, Kolmogorov–Smirnov test $P < 0.0001$; ipsilateral vs. spontaneous; Kolmogorov–Smirnov test $P < 0.0001$, Fig. 3f, $N = 30$ neurons, $N = 5$ populations with six neurons each). Upon inactivation of the vc pathway, population spiking significantly reduced for all stimuli indicating that for small populations of visually responsive neurons, the vc pathway contributes significantly to the generation of large population-wide spiking events for both the ipsilateral and contralateral visual pathways (Fig. 3f, Kolmogorov–Smirnov test: binocular $P < 0.0001$, contralateral $P < 0.0001$, ipsilateral $P < 0.0001$). As bursting activity from a few neurons could strongly influence these statistics, we next

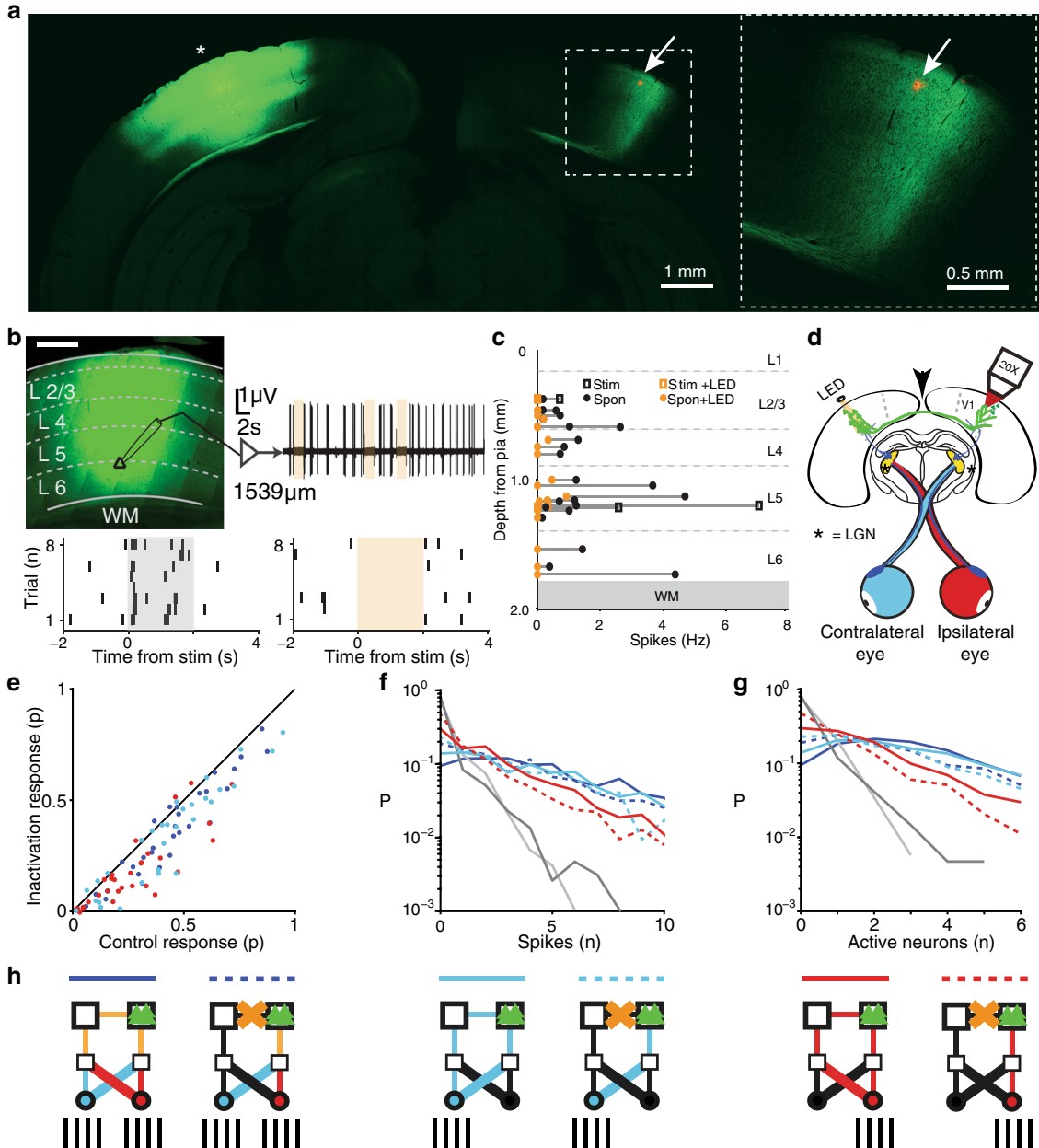

**Fig. 3 Anterograde tracing of callosal projection neurons and effects of optogenetic inhibition. a** Coronal section of AAV-YFP injection into binocular V1 (asterisk). Right, enlarged view of region outlined by box in left panel. Note the red dye injection site for anatomical identification (arrow, same as in right). **b** Coronal V1 section expressing eArchT-YFP (left, scale bar 0.5 mm) and example cell-attached recording from a neuron 1539 µm below pial surface during periods of spontaneous activity without and with eArchT activation (upper, orange boxes). (Lower) Raster plots from a cell-attached recording showing responses to visual stimuli without (left) and with eArchT activation (right, orange box). **c** Pooled data of firing rates before (black) and during activation of eArchT (orange) with recording depth for both spontaneous activity (circles) and stimulus-evoked activity (squares). **d** Schematic showing ipsilateral population imaging site (objective lens), location of contralateral callosal projecting neurons (arrow), ipsilateral (red) and contralateral (turquoise) eye crossed optical pathway targeting lateral geniculate nuclei (LGN, star). Uncrossed pathways in blue. **e** Average response probabilities from populations of visually responsive neurons (6 populations with 41 neurons) during binocular (blue), contralateral (turquoise), and ipsilateral stimulation (red) to moving gratings across all eight angles with and without inactivation of vc pathway. **f** Probability distributions of number of inferred action potentials evoked per stimulus period in a set population of visually responsive neurons (five populations with six neurons each) during binocular (blue), contralateral (turquoise), ipsilateral (red) stimulation, spontaneous (dark gray), and shuffled (Poisson) spontaneous (light gray). Dashed lines indicate inactivation of vc pathway. **g** Probability distributions of number of active neurons per stimulus period. Populations and line conventions as in **f**. **h** Diagrams depicting visual pathways activated during different conditions in **f**, **g**; unbroken lines control, dashed lines vc pathway inactivation. Direct (turquoise, red, vertical) and crossed (turquoise, red, diagonal) visual pathways, LGN (small square), contralateral (large open square) and ipsilateral (large square, green triangles) cortex, with callosal pathway (line between large squares) intact or inactivated (yellow cross). Left panel: Binocular stimulation (lower bars). Middle panel: Contralateral stimulation. Right panel: Ipsilateral stimulation.

compared the spiking probability distributions based on the number of active neurons per stimulus trial ($N = 5$ populations with six neurons each, Fig. 3g). Both binocular and contralateral stimulation evoked similar probability distributions of active neurons per stimulus, whereas ipsilateral stimulation activated significantly fewer neurons (Fig. 3g; binocular vs. contralateral, Kolmogorov–Smirnov test $P = 0.8532$, excluding 0 bin; binocular vs. ipsilateral; Kolmogorov–Smirnov test $P < 0.0001$, excluding 0 bin). Binocular stimulation had the highest probability of evoking one or more neurons per stimulus trial compared to monocular stimulation of either eye (probability of 0 neurons active: binocular: 0.0942, contralateral: 0.1386, ipsilateral: 0.2983). For both binocular and monocular stimuli, vc pathway inactivation significantly increased the number of trials where 0 neurons were active and also significantly decreased the probability of observing activity in groups of neurons of all sizes (binocular LED off vs. on, Kolmogorov–Smirnov test $P < 0.0001$; contralateral LED off vs. on, Kolmogorov–Smirnov test $P < 0.001$; ipsilateral LED off vs. on, Kolmogorov–Smirnov test $P < 0.0001$). Together, this shows that both the crossed and uncrossed visual pathways originating from either eye utilize the vc pathway (Fig. 3h). We next quantified how this pathway is involved in the suprathreshold responses of individual neurons and whether individual response classifications rely on this pathway.

**Response amplitude is altered but not preferred orientation.** To quantify effects on orientation tuning, tuning curves were estimated by fitting Gaussian curves to the raw orientation responses[29,30], and the peak angle of the Gaussian fits were compared between control and vc pathway silencing conditions. From all tuned neurons (101 neurons, $N = 17$ animals), 31% significantly reduced response amplitudes upon silencing the vc pathway, with 11% of these neurons becoming non-responsive (not significantly different from spontaneous activity; excluded from further statistical analysis). While the response amplitudes at the preferred orientation were significantly reduced (Gauss. fitted amplitude, control vs. vc pathway inactivated, $-1.23 \pm 0.69$ Hz, mean $\pm$ SD, Wilcoxon signed-rank test $P < 0.0001$, $N = 23$ neurons, Supplementary Figs. 6a, 7 and 8), the preferred orientation was not significantly changed (Supplementary Fig. 6b, absolute difference in preferred orientation: $7.85 \pm 7.94°$, mean $\pm$ SD, circular one-sample $t$-test, $P > 0.1$, $N = 23$ neurons). In summary, while the vc pathway strongly influences stimulus response strength in a subpopulation of neurons, it has no significant influence on orientation tuning.

**Vc pathway alters responses in monocular and binocular neurons.** We next asked whether individual monocular and binocular neurons rely on this pathway for their response properties. We first quantified how the activity of individual neurons, classified as either monocularly responsive (ipsilateral or contralateral) or binocularly responsive (Fig. 1g), was modulated by the vc pathway (Fig. 4a). Silencing significantly changed stimulus-evoked firing rates in 30.61% of visually responsive neurons (45 from 147 responsive neurons, total of 279 neurons recorded, $N = 17$ animals, see "Methods" section for details). This change occurred in both classes of monocularly responsive neurons, as well as in binocularly responsive neurons (contra-responsive 14/63, ipsi-responsive 5/28, binocular 26/56 neurons, Fig. 4a). This modulation was not associated with a change in the spontaneous firing rates between the two conditions (before vs. after contralateral silencing, $0.092 \pm 0.25$ vs. $0.080 \pm 0.27$ Hz, median $\pm$ SD, two-sample $t$-test, $P = 0.704$).

From this analysis, it was clear that vc pathway inactivation had a range of effects on stimulus-evoked spiking, from ineffective

(Fig. 4a, b) to reducing visual stimulus responses to not significantly different to spontaneous activity (Fig. 4c, Supplementary Fig. 9). Stimulus-driven activity from each eye has two potential visual pathways (crossed vs. uncrossed pathway) to V1, of which one involves the vc pathway (Fig. 4d). One possibility was that the wide range of effects of vc pathway inactivation was due to functionally heterogeneous inputs driving the neurons arising from the different visual pathways. As both of the monocular pathways can give rise to either the binocular neurons' spiking responses or monocular neurons' spiking responses, we next separately quantified the contribution of each pathway for both monocular and binocular neurons to establish which of the pathways was most affected by vc inactivation (Fig. 4d).

First, we ranked all visually responsive neurons according to the change in their mean stimulus-evoked firing rate caused by vc pathway inactivation (45 neurons significantly modulated, 26 binocular, 5 ipsilateral, and 14 contralateral (Fig. 4e), same data as in Fig. 4a), and concurrently quantified for each neuron which of the two monocular visual pathways was effected the most (Fig. 4e, color coding denotes effect mainly on contralateral (more turquoise) or ipsilateral (more red) eye response, black arrow denotes example in Fig. 4c, gray arrow denotes example in Supplementary Fig. 10). This analysis showed that the ipsilateral crossed pathway (ipsi-eye→contra-LGN→contra-V1→ipsi-V1 via vc) reduced most in binocular neurons, especially in neurons where responses after vc pathway inactivation were indistinguishable from spontaneous rates (Fig. 4c, e and Supplementary Fig. 11a, ipsilateral vs. contralateral stimulus response reduction, $54.08 \pm 26.88\%$ vs. $35.95 \pm 29.51\%$, mean $\pm$ SD, Mann–Whitney $U$ test $P = 0.036$). For this subpopulation of both monocularly and binocularly responsive neurons, inactivation of the callosal input resulted in an almost complete cessation of responses to ipsilateral and contralateral eye stimulations (10 binocularly responsive, 2 ipsilaterally responsive, and 2 contralaterally responsive (Fig. 4e), Kruskal–Wallis nonparametric ANOVA followed by one-tailed Fisher's test, $P > 0.05$). It could be expected that the contribution from the vc pathway forms a continuum from suprathreshold responses being completely independent of the vc pathway to them being completely reliant on activation of the vc pathway (Fig. 4f). We next determined the fraction of neurons in these different vc pathway-dependent classes.

**Neurons can be dependent or independent of vc pathway.** For a binocular neuron, a reduction in responsiveness upon stimulation of the ipsilateral eye by inactivating the vc pathway implies that the pathway must involve the crossed projection through the contralateral LGN to contralateral V1, and then the vc pathway from contralateral V1 (Fig. 4f, right panel). It follows that a reduction in responsiveness upon stimulation of the contralateral eye implies that the pathway must involve the uncrossed projection from the contralateral eye through the contralateral LGN to contralateral V1 and then the vc pathway from contralateral V1 (Fig. 4f, right panel). By quantifying the relative reduction of each of the monocular pathways during vc pathway inactivation compared to controls, each binocular neuron's reliance on the vc pathway could be compared. This analysis showed that for these binocular neurons the contribution made by the vc pathway can be via either of the monocular (ipsilateral or contralateral) pathways or both (Fig. 4e, circular markers colored denoting reduced pathway), suggesting that visually driven suprathreshold activity can be generated from inputs arising from any of the potential visual pathways that innervate VCPNs. This was not specific to binocular neurons, as using the same analysis approach for monocular neurons revealed a similar range of responses to vc pathway inactivation (Fig. 4e). For both contralaterally and

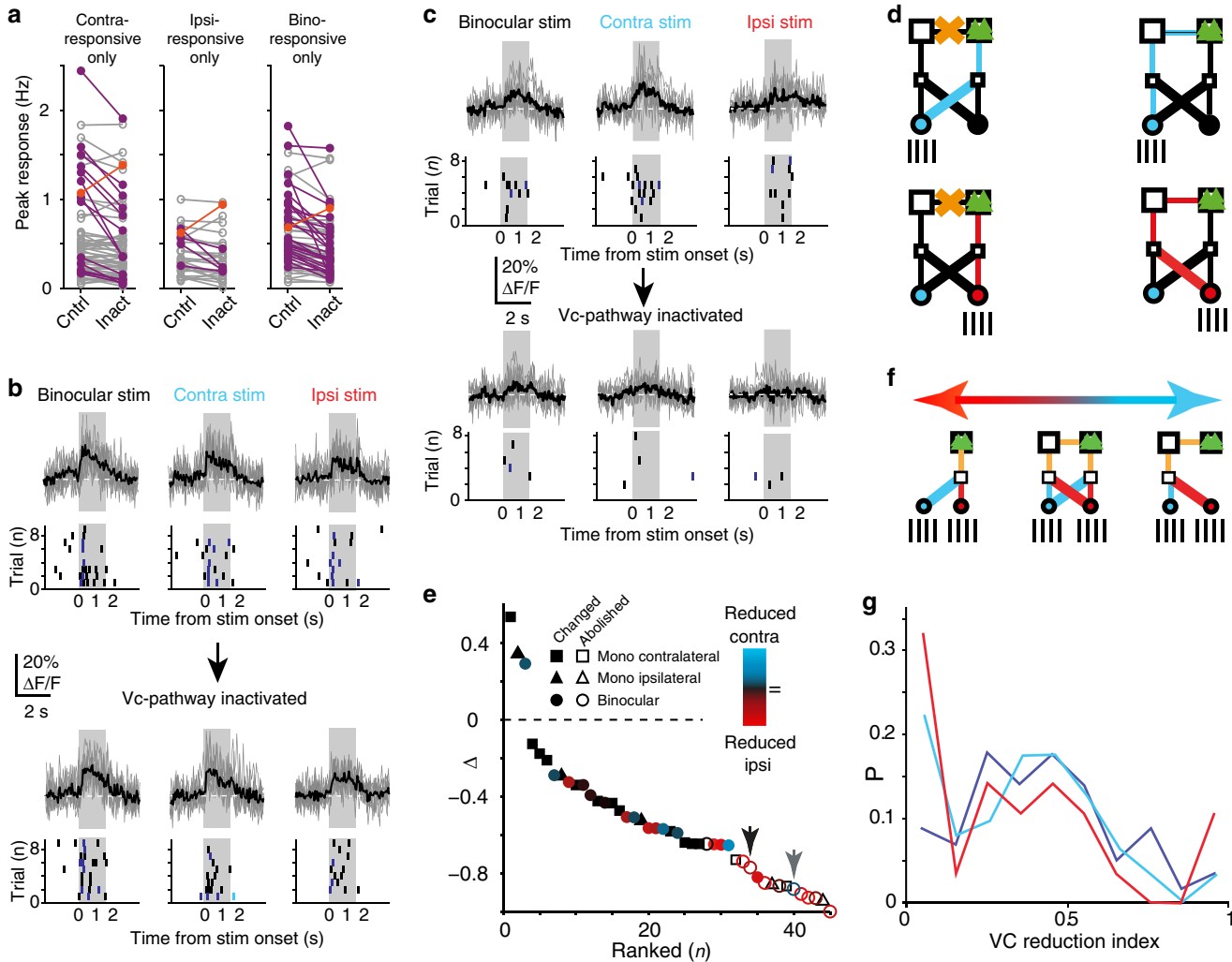

**Fig. 4 Callosal pathway modulates spiking responses in binocular and monocular neurons. a** Firing rate changes for maximum response angle from visually responsive neurons without (control, Cntrl) and with (Inact.) callosal pathway inactivation for contralaterally only (left), ipsilaterally only (middle) and binocularly only (right) responsive neurons. Significant (purple, orange) and non-significant (gray) changes depicted. **b** Example responses from a neuron not influenced by vc pathway inactivation. Upper panel, example of $Ca^{2+}$-transients from each stimulus trial (gray) and resulting average (black) during binocular (left), contralateral (middle), and ipsilateral eye stimulation (right) for maximum response angle (see "Methods" section), inferred spikes (singles black, doubles dark blue, triples light blue) from transients for each trial (8 trials). Lower panel, same as in upper panel, but with vc pathway inactivation. **c** Example responses from a neuron showing a strong vc pathway inactivation effect. Convention same as in **b**. **d** Visual pathways activated during control and vc inactivation conditions, conventions as in Fig. 3h. **e** Graph depicting amount of significant firing rate change in binocularly (circles), ipsilaterally (triangles), and contralaterally (squares) responsive neurons during silencing of the callosal pathway (same as purple and orange in **a**). Neurons ranked from those that increased firing rates to those that decreased firing to not significantly different from spontaneous activity (open symbols). Symbol color denotes whether ipsilateral (red) or contralateral (turquoise) response decreased most or both decreased equally (black). Black arrow depicts example neuron shown in **c**, gray arrow depicts example neuron shown in Supplementary Fig. 10. **f** Visual pathways activated during binocular stimulation, showing different amounts of contribution of the vc pathway, ranging from responses being completely reliant on (right) to being independent of (left) vc pathway activation. **g** Probability distribution of the amount of influence of the vc pathway on responses, quantified as the vc reduction index for binocularly (blue), ipsilaterally (red), and contralaterally (turquoise) responsive neurons.

ipsilaterally responsive neurons, responses to visual stimuli at the preferred orientation were reduced to spontaneous rates for a small number of neurons ($N = 2$ contralaterally responsive neurons, $N = 2$ ipsilaterally responsive neurons). To more directly compare the contribution from the vc pathway for monocular and binocular neurons, we calculated the vc reduction index. This index was bounded between 0 and 1, with 0 denoting no contribution from the vc pathway and 1 denoting complete reliance on the vc pathway (see "Methods" section for details). As for the binocular neurons, the majority of neurons' monocularly driven suprathreshold activity arose from a combination of crossed and uncrossed pathway activation, with a small number of neurons'

suprathreshold activity unaffected by vc inactivation (Fig. 4g). This is consistent with the presence of representatives of all of the potential anatomical pathways from the eye to the VCPNs within the total population of monocular neurons in V1. In addition, although callosally projecting axons have been observed to accumulate into patches along the lateral edge of V1 in flattened-cortex histological preparations from rats[11], we did not observe any clustering of neurons whose responses were more strongly modulated by vc pathway inactivation (Supplementary Fig. 11b). The above analyses suggest that the vc pathway plays a crucial role in generating spiking responses within its innervation domain in binocular primary visual cortex, but this does not

necessarily imply that there are neurons wholly reliant on this pathway for their inputs or subthreshold activity.

**Visual responses are abolished by silencing ipsilateral LGN.** We next investigated whether the vc pathway alone was able to generate spiking activity (Fig. 5a–c). We inactivated the LGN on one side with muscimol (8.8 mM) and recorded visually evoked responses in V1 on the same side ($N = 4$ animals, $N = 28$ visually responsive neurons). In a subset of these experiments ($N = 2$ animals, $N = 21$ visually responsive neurons), we reversibly inactivated the vc pathway before LGN inactivation to confirm that vc inactivation had the same effect as described before (compare Figs. 5d and 4a). Vc pathway inactivation modulated visually responsive neurons with the same average reduction as in the previous experiments (Fig. 5d, Kolmogorov–Smirnov test $P = 0.4741$, Supplementary Fig. 12). As in the previous experimental group, a subpopulation of neurons were observed whose responses to stimulation of the ipsilateral eye were abolished on vc pathway silencing (Fig. 5d, $N = 2$ neurons). The inactivation of ipsilateral LGN abolished visually evoked spiking responses in all visually responsive neurons ($N = 4$ animals, $N = 28$), including neurons where responses had been reduced to the level of spontaneous activity on vc pathway silencing. Firing rates in neurons responsive to ipsilateral eye stimulation were significantly reduced upon LGN inactivation ($1.585 \pm 0.79$ Hz vs. $0.217 \pm 0.108$ Hz, mean $\pm$ SD, paired Wilcoxon signed-rank test $P < 0.0001$, $N = 10$ binocular neurons and five ipsi-monocular neurons) and in the order of spontaneous rates ($0.301 \pm 0.135$ Hz, mean $\pm$ SD, spontaneous rates before LGN inactivation: paired Wilcoxon signed-rank test $P = 0.018$; $0.101 \pm 0.09$ Hz, mean $\pm$ SD, spontaneous rates after LGN inactivation: paired Wilcoxon signed-rank test $P = 0.0026$). Also, firing rates in neurons responsive to contralateral eye stimulation were significantly reduced upon LGN inactivation ($1.635 \pm 0.779$ Hz vs. $0.206 \pm 0.185$ Hz, mean $\pm$ SD, paired Wilcoxon signed-rank test $P < 0.0001$, $N = 10$ binocular neurons and 13 contra-monocular neurons) and not different to spontaneous firing rates ($0.243 \pm 0.136$ Hz, mean $\pm$ SD, before LGN inactivation, paired Wilcoxon signed-rank test $P = 0.307$; $0.157 \pm 0.123$ Hz, mean $\pm$ SD, after LGN inactivation, paired Wilcoxon signed-rank test $P = 0.2734$). Responses prior to inactivation of LGN compared to responses after inactivation of LGN are shown for eight representative binocular neurons (Fig. 5d, ipsilateral eye stimulation: $1.778 \pm 0.703$ Hz vs. $0.258 \pm 0.094$ Hz, mean $\pm$ SD, paired Wilcoxon signed-rank test $P = 0.0078$; contralateral eye stimulation: $2.068 \pm 1.044$ Hz vs. $0.24 \pm 0.158$ Hz, mean $\pm$ SD, paired Wilcoxon signed-rank test $P = 0.0078$).

To confirm that muscimol injection was confined to ipsilateral LGN and not causing a more general inactivation of visual responses, we examined neuropil-related Ca$^{2+}$-fluorescence signals, which should include fluorescence transients in the vc pathway axons. Bulk loading of cortical tissue is known to label the neuropil in addition to neurons and astrocytes within the loaded area. Labeled neuropil also displays strong fluorescence intensity fluctuations, which have been shown to be correlated with both neuronal intracellular membrane potential fluctuations and the electrocorticogram[31,32]. Furthermore, the neuropil-related Ca$^{2+}$-signals have also been shown to originate from labeled axonal structures[31]. After muscimol injection, visual stimuli to either the ipsilateral or contralateral eye evoked small amplitude Ca$^{2+}$-transients in the neuropil (Fig. 5e upper). These transients were locked to the visual stimuli, consistent with the stimulus evoking activity in axons whose somata of origin lay outside of ipsilateral LGN. These neuropil transients were not found in a shuffled-data control of

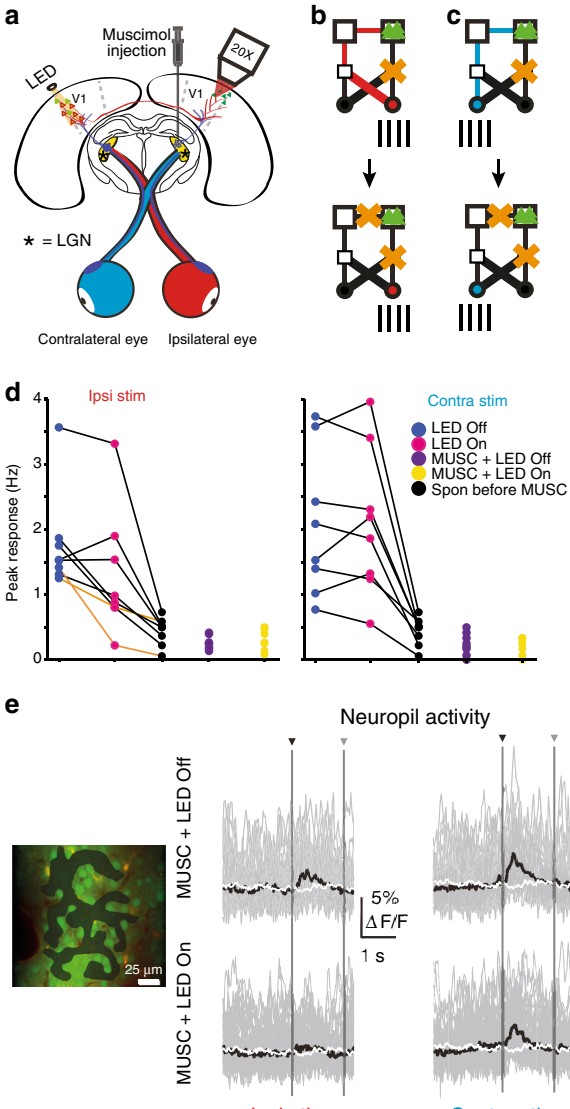

**Fig. 5 Activation of callosal pathway alone from either eye is not sufficient to drive spiking.** **a** Schematic of experimental design showing inactivation of ipsilateral LGN (muscimol injection) and reversible inactivation of vc pathway. **b** Cartoon depicting pathways deactivated during muscimol injection (upper) for ipsilateral stimulation and for experiments with muscimol injection and vc pathway inactivation (lower). **c** Cartoon depicting pathways deactivated during muscimol injection (upper) for contralateral stimulation and for experiments with muscimol injection and vc pathway inactivation (lower). **d** Comparison of spiking response for maximum response angle during ipsilateral (left, 8 neurons, ipsilaterally responsive, 2 animals) and contralateral (right, 8 neurons, contralaterally responsive, 2 animals) only stimulation. Control (dark blue, no muscimol), vc pathway inactivation (magenta), LGN inactivation (purple), and LGN inactivation with vc pathway inactivation (yellow). Spontaneous activity rates (black) shown for each neuron. Ipsilateral responses not different to spontaneous activity on vc pathway silencing indicated by orange lines. **e** Left panel, example field of view with labeled neuropil ROI (shaded area); right panel, neuropil responses evoked by stimulation of ipsilateral (left) or contralateral (right) eye recorded during silencing of ipsilateral LGN without (upper) and with VCPN-inactivation (lower). Individual (gray) and average (black) responses. Shuffled neuropil responses (white). Stimulus onset (black arrow head) and offset (gray arrow head).

this analysis (Fig. 5e, white traces, two-sample $t$-test $P = 0.304$). Optogenetic silencing of contralateral hemisphere, and therefore the vc pathway axons, abolished the neuropil $Ca^{2+}$-transients evoked by stimulation of ipsilateral eye ($Ca^{2+}$-transients not significantly different from shuffled data, two-sample $t$-test $P = 0.691$, Fig. 5e, left lower). Callosal silencing had little effect on the neuropil-related $Ca^{2+}$-transients evoked by stimulation of the contralateral eye, suggesting that the contralateral eye has a projection that does not involve the vc pathway or the projection from the ipsilateral LGN ($Ca^{2+}$-transients not significantly different between silencing and non-silencing conditions, two-sample $t$-test $P = 0.3261$ (Fig. 5e, right lower). This strongly suggests that the vc pathway provides stimulus-driven activity that is alone unable to drive the ipsilateral neurons to spike but spiking activity is contingent upon co-activation of the ipsilateral LGN.

## Discussion

Here, we show that neuronal populations in lateral binocular V1 contain a mixture of projecting, i.e. VCPNs, and non-projecting neurons, non-VCPNs, that do not show obvious spatial organization. Functionally, VCPNs can be binocularly or monocularly responsive to stimulation of either ipsilateral or contralateral eye and, taken as a population, their functional tuning is not significantly different from non-VCPNs. Upon inactivation of the callosal pathway, spiking responses in a large proportion of contralateral V1 neurons were modulated, both monocularly and binocularly responding. The effect of vc pathway inactivation was marked, and many neurons showed a significant spiking reduction or, in some cases, a complete extinction of responses to visual stimuli. We also show that a subpopulation of neurons depends on both input from callosal projection neurons and simultaneous input from the ipsilateral LGN for their binocular spiking responses.

Studies using retrograde tracing or virus-mediated expression techniques are limited at least to some extent by the total fraction of the target neuronal population labeled. In the current study, this applies both to the combined retrograde tracing and $Ca^{2+}$-imaging experiments as well as to the optogenetic manipulation of contralateral V1. In retrograde tracing experiments, we found that around 35% of neurons in layer 2/3 were labeled. Compared to the unlabeled neurons, the labeled population did not show significant differences in tuning and response properties, though the possibility still remains that any unlabeled projection neurons may have more selective visual response properties. A similar possibility exists also for the optogenetic manipulation experiments. However, given that an appreciable fraction of neurons show significant modulation of responses, including reduction of evoked responses to the level of spontaneous activity, we clearly show that activity in callosally projecting neurons is an important component of visual responses for many V1 neurons.

The callosal projection in the visual cortex has been observed in many mammalian species[4–10,33–35]. Functionally, initial studies in cats showed that the callosal projection targets the retinotopically matched region of contralateral visual cortex[3,4] and, similar to the findings in the current study, that inhibition of CPNs by cortical cooling has a variety of consequences for the visual responses of neurons in the recipient cortical region including a general reduction in activity[36]. One key finding in the current study is the callosal projection in rats is only able to drive spiking responses in their projection targets when there is simultaneous activity in the pathway from the ipsilateral LGN. Given that there are broad similarities in callosal anatomy and function across the different species described above it seems

possible that this might also be the case for the callosal projection in other species too.

Previous studies investigating the role of the vc pathway in rodents are contradictory, some concluding that the callosal pathway contributes ipsilateral-eye-derived visually evoked activity[13,22,23], with others concluding that it does not[11,24]. Also, some previous studies found no effect of callosal pathway inactivation on contralateral eye responses[23]. Our current findings showing that callosal pathway inactivation has a wide range of effects on cortical spiking, from no-effect to complete spike abatement, go some way to reconcile the differences between seemingly contradictory findings of previous studies. This is most likely because the present study was able to record simultaneously from populations of visually responsive neurons across large spatial areas and reversibly inactivate VCPN somas across all cortical layers.

In the current study, we found no significant influence of inactivation of the vc pathway on orientation-tuning properties of V1 neurons. For some neurons, strongly orientation tuned responses were no longer present in the presence of vc pathway inactivation (see Supplementary Figs. 7, 8), suggesting that the orientation tuning was due to the orientation tuning of the pre-synaptic callosally projecting neurons. For other neurons, strongly tuned responses were reduced in response amplitude, but without changing the preferred orientation (see Supplementary Figs. 7, 8). These findings raise the possibility that the axons of neurons projecting through the vc pathway target contralateral neurons with similar orientation preference to that of the projecting neurons, consistent with the findings of a recent study showing that spines in V1 neurons connecting with axons from the vc pathway cluster more with spines from local neurons with similar orientation preference[25]. This same study also showed that the callosal-recipient spines have more similar orientation preference to the soma than non-callosal recipient spines[25].

In the LGN inactivation experiments, the presence of a visually evoked neuropil signal is an important indication that the contralateral LGN and callosal pathway are still viable and active in the presence of ipsilateral LGN inactivation. As the neuropil signal represents $Ca^{2+}$-influx into synaptic boutons on axons labeled with $Ca^{2+}$-indicators[31], this signal probably represents the activity in the axons of the VCPNs. The finding that the neuropil signal remaining during VCPN inactivation is abolished by inactivation of the ipsilateral LGN (Fig. 5) is consistent with there being only two possible pathways from ipsilateral eye to visual cortex, the uncrossed projection from the ipsilateral retina and the crossed projection to the contralateral LGN and visual cortex via the callosal projection (ipsilateral eye–contralateral LGN–contralateral visual cortex–ipsilateral visual cortex via corpus callosum). However, as the neuropil response to contralateral eye stimulation was not abolished with the combination of VCPN silencing and LGN inactivation, it would seem that there is an additional anatomical pathway activated by contralateral eye stimulation. One potential pathway is the retinal projection to the lateral posterior nucleus of the thalamus, and the latter's projection to V1 (refs. [2,37]).

In conclusion, we have shown that the visual callosal pathway, while contributing substantial input to many neurons and being responsible for all spiking responses in some neurons, requires concurrent activation of inputs from the ipsilateral thalamus in order to drive suprathreshold responses. This suggests a much greater role of the rodent callosal pathway in cortical processing than previously thought.

## Methods

All experiments were carried out on male Lister hooded rats (*Rattus norvegicus*). For retrograde anatomical tracing experiments, rats ($N = 5$) were between 93 and

160 g body weight at the time of tracer injection. For anterograde axonal tracing experiments, rats ($N = 2$) were of 55 and 58 g body weights at the time of tracer injection. For experiments combining imaging and retrograde tracing, rats ($N = 3$) were between 135 and 142 g body weight at the time of tracer injection. For experiments involving imaging or electrophysiology together with V1 inactivation and/or LGN inactivation, rats ($N = 31$) were between 137 and 287 g body weight at the time of the recordings. Experiments were approved by the relevant animal welfare authorities (Regierungspraesidium Tuebingen and Landesamt für Natur, Umwelt und Verbraucherschutz Nordrhein-Westfalen, Germany).

**Animal preparation for anatomical tracing experiments.** Animals were anesthetized with ketamine (200 µg/kg)/medetomidine (100 mg/kg), and body temperature maintained at 37 °C using a thermal probe and heating pad. Depth of anesthesia was monitored throughout the procedure via assessment of withdrawal reflexes with supplemental doses of anesthetic provided where necessary. The head was immobilized using ear and tooth bars and the skin and galea on the dorsal aspect of the skull opened. For mono-color retrograde tracing experiments, a small (~400 × 400 µm) craniotomy was opened over the right visual cortex (mean ± SD; lambda −2.3 ± 0.2 mm, lateral 3.9 ± 0.2 mm), the underlying dura opened and 115 nL cholera toxin subunit B (CTB) conjugated to Alexa 594 (Molecular Probes, OR, USA) injected using a custom-built injection setup. For tri-color retrograde tracing experiments, three small (~400 × 400 µm) craniotomies were opened over the left visual cortex (rat 1; lambda −1.6, −1.7, and −1.6 mm, lateral 2.0, 2.8 and 4.0 mm, rat 2; lambda −1.5, −1.4, and −1.6 mm, lateral 2.9, 3.4, and 4.3 mm), small openings in the underlying dura made in each and one of either CTB conjugated to Alexa Fluor 488, 594, or 647 (all from Molecular Probes, OR, USA) injected into each craniotomy at 0.45 and 1.05 mm below the pia (46 nL at each depth). For anterograde tracing, a single craniotomy (~400 × 400 µm) was made over the left visual cortex (rat 1; lambda −1.0 mm, lateral 4.1 mm, rat 2; lambda −1.0 mm, lateral 4.1 mm), a small opening in the dura made, and an injection of AAV5-CaMKIIα-eArchT3.0-eYFP (UNC Vector Core, NC, USA) that expresses yellow fluorescent protein was made (injection depth from pia 0.45 and 1.15 mm, 184 nL at each depth). Following the injections, the craniotomy was protected with KwikSil (World Precision Instruments, FL, USA), and the skin incision closed with vicryl sutures. 14–19 days after CTB injections or 20–22 days after anterograde tracer injections, animals were deeply anesthetized, perfused transcardially with 4% paraformaldehyde (Roti-histofix 4%, Carl Roth GmbH, Karlsruhe, Germany) and the brain removed. After postfixation for at least 12 h, 100 µm-thick coronal sections were cut on a vibratome or a freezing-microtome and mounted using anti-fade mounting medium (Vectashield, Biozol Diagnostica Vertrieb GmbH, Eching, Germany). Images of the coronal sections and fluorescently labeled neurons were subsequently acquired on a conventional fluorescence microscope or confocal microscope.

**Animal preparation for anatomical tracing and Ca$^{2+}$-imaging.** For experiments combining Ca$^{2+}$-imaging and retrograde tracing of contralateral projection neurons, an injection of Alexa Fluor 594 conjugated CTB was made in the right hemisphere at lambda −1.7 ± 0.1 mm, lateral 4.3 ± 0.1 mm (mean ± SD) as described above. After 10–15 days, animals were anesthetized with urethane (1.9 g/kg) and prepared for in vivo Ca$^{2+}$-imaging. The skin and galea on the dorsal aspect of the skull were removed and a custom-made metal headplate implanted over the left visual cortex (contralateral to the previous CTB injection). A 2 × 2 mm craniotomy was made, and the underlying dura removed. The exposed cortex was then stabilized with agar (1.2%, Sigma Aldrich, MO, USA) and a coverslip. Body temperature was maintained at 37 °C throughout the experiment using a thermal probe and heating pad, and anesthetic depth monitored throughout via assessment of withdrawal reflexes with additional anesthetic doses provided where required.

**Animal preparation for cortical silencing and neural recording.** Animals were anesthetized and injected with AAV-eArchT as described above for anatomical tracing experiments, with the craniotomy (~400 × 400 µm) made over either left or right visual cortex at lambda −1.3 ± 0.2 mm, lateral 4.1 ± 0.2 mm (mean ± SD). Expression time after injections was between 15 and 24 days. For Ca$^{2+}$-imaging experiments, animals were anesthetized with urethane (1.9 g/kg), the skin and galea on the dorsal aspect of the skull removed, a custom-made metal headplate implanted over either left or right visual cortex contralateral to the previous AAV-eArchT injection site, a 2 × 2 mm craniotomy opened and the underlying dura removed. The underlying cortex was then stabilized with agar and a coverslip. A craniotomy (3 × 3 mm) was also opened over the previous AAV-eArchT injection site and a LED (Golden Dragon, 590 nm, OSRAM, Regensburg, Germany) with attached half-ball lens (S-LAH79, 2 mm diameter, Edmund Optics, NJ, USA) was mounted over the cortical site, with shielding to prevent the light from escaping the cortex. The effective power density of the LED at the cortical surface was 12.8 mW/mm$^2$. For electrophysiology experiments to confirm the effectiveness of eArchT-induced neuronal silencing, animals were prepared as described for imaging experiments with the exception that the craniotomy (3 × 3 mm) was opened only over the site of previous AAV-eArchT injection in either left/right visual cortex.

**Animal preparation for cortical and thalamic silencing.** The procedure for animal preparation was similar to that described above for combined cortical

silencing with eArchT and Ca$^{2+}$-imaging, with the difference that an additional craniotomy (1 × 1 mm located at Lambda −3 mm, lateral 2 mm) was opened in the same hemisphere in which Ca$^{2+}$-imaging was carried out. Through this craniotomy after the completion of the first session involving simultaneous cortical silencing and Ca$^{2+}$-imaging, a glass pipette was advanced 3.6 mm into thalamus (LGN) and a 184 nL bolus of 8.8 mM muscimol (Sigma Aldrich, MO, USA) injected, after which a second session of Ca$^{2+}$-imaging was carried out.

**Electrophysiological recording.** Electrophysiological recordings were made using glass pipettes filled with artificial cerebrospinal fluid (ACSF) with the following composition (in mM): 135 NaCl, 5.4 KCl, 1.8 CaCl$_2$, 1 MgCl$_2$, 5 HEPES, pH balanced to 7.2 (300 mOsm/l). Signals were amplified using a multiclamp 700B amplifier (Molecular Devices, CA, USA), and digitized at 31.25 kHz using a Power 1401 analog to digital (AD) converter (Cambridge Electronic Design, Cambridge, UK). Timing of visual stimuli was recorded using the same AD converter to allow analysis of stimulus-related action potential firing.

**Intrinsic imaging of visually related brain activity.** Intrinsic imaging was performed as described previously[38]. In brief, excitation wavelength was 630 ± 30 nm. Visual stimulation consisted of full-field rectangular drifting gratings (width × height; 45° × 35°, displayed 48 cm in front of rat's eyes, 8 drift directions, other stimulus properties as described below) presented for 5 s with inter-stimulus interval of 30 s. Image acquisition started 2 s before the onset of each stimulus presentation and ended 3 s after the offset of stimulus presentation (images were acquired every 100 ms within this period). The images acquired before the onset of stimulus presentation were averaged together, with the result used for background subtraction. Image frames between stimulus onset and offset were averaged and a contour at the 90 percentile used as indicative of the region activated by the visual stimulus. To facilitate navigation under multiphoton microscopy and to allow subsequent alignment of multiphoton data from multiple animals on the basis of the intrinsic optical signal imaging response (see below for details), we also acquired an image of the cortical surface vasculature.

**Ca$^{2+}$-indicator loading and astrocyte counterstaining.** Cortical layer 2/3 neurons were loaded with the Ca$^{2+}$-indicator Oregon green 488 BAPTA-1 AM (Thermo Fischer, MA, USA) and astrocytes with sulforhodamine 101 (Sigma Aldrich, MO, USA) as described previously[39,40].

**Multiphoton microscopy.** Multiphoton imaging was performed using a custom-built, laser scanning multiphoton microscope as described in refs. [39,41]. In a subset of experiments, the galvanometric scanners were replaced with a resonant scanning system consisting of an 8 kHz resonant scanner on one scan axis (Sutter Instruments, CA, USA) and conventional galvanometric scanner on the other. Excitation was provided by a Mai Tai laser (Spectra Physics, CA, USA) at 920 nm and images acquired using Scanimage software (Vidrio Technologies, VA, USA). The objective lens used for all functional imaging was a Zeiss ×20 1.0 NA objective (Carl Zeiss AG, Oberkochen, Germany). In experiments using the conventional galvanometric scanning system, 128 × 64 pixel full frame images were acquired at a frame rate of 18.6 Hz, and in those using the resonant scanning system, 512 × 512 pixel full frame images were acquired at 29.98 Hz.

**Visual stimulation.** All visual stimuli were created, controlled, and displayed using Psychtoolbox software[42] executed in Matlab (Mathworks, MA, USA). Visual stimuli were presented on a CRT monitor (Sony CPD-G520, resolution: 1600 × 1200, refresh rate: 60 Hz) having constant gray background (30 candela/m$^2$) that changed to a full-field stimulus image (mean luminance: 30 candela/m$^2$). The CRT monitor was placed 48 cm in front of eyes and occupied 45° × 35° (width × height) in visual space centered in front of the animal's nose.

Rectangular drifting grating stimuli (spatial frequency 0.05 cycle/degree, drift speed 40 degrees/s, contrast 96.72%, duration 2.0 s in experiments combining Ca$^{2+}$-imaging with CTB tracing and duration 1.5 s in experiments combining Ca$^{2+}$-imaging with cortical silencing) were presented at eight angular orientations and two directions to give 16 separate grating stimuli in total. In addition, a blank stimulus (same as gray background) of same duration as grating stimuli was presented, that served as a measure of spontaneous activity. The stimulus presentations were pseudo-randomized. Automated shutters were used to exclude visual stimuli from one eye or the other in order to deliver monocular stimuli, and each grating was presented a total of 8 times to each eye monocularly, and in a subset of 5 animals also 8 times to both eyes (binocularly). The automated shutters were custom made, and consisted of a neoprene cup to cover the eye connected via a 8 cm long rod to a servo motor (Supplementary Fig. 13). The neoprene cup was placed over the eye, pressing slightly into the surrounding fur, and could be retracted at an angle downwards away from the eye using the servo motor to allow visual stimulus presentation. The servo motor position was controlled via the visual stimulus software to allow full computer control and randomization of monocular or binocular stimulus presentation. Inter-stimulus interval was between 2 and 3 s (randomized across trials) in experiments combining Ca$^{2+}$-imaging and CTB tracing and 3–4 s (randomized across trials) in experiments combining Ca$^{2+}$-imaging and cortical silencing. In

experiments combining Ca²⁺-imaging, cortical silencing, and thalamic silencing, the rectangular drifting grating stimuli with only one drift direction were used, otherwise all other stimuli parameters remained the same as that in experiments combining Ca²⁺-imaging and cortical silencing.

**Data analysis**. Analyses were performed using custom-written routines executed in Matlab (Mathworks, MA, USA). For quantification of CTB labeling density across animals, the coronal sections of brains injected with CTB-594 (CTB conjugated to Alexa 594) were used. The images of coronal sections were acquired using a fixed exposure time. From the images of coronal sections, the 3D brain volume was reconstructed and the reconstructed brain volume from each animal was aligned to rat brain atlas[43]. The mean CTB labeling density from all animals was then quantified in the co-ordinates of rat brain atlas. This process was carried out through manually assisted custom-written routine in Matlab and is briefly as follows; the images of each coronal section were split along the midline and the split images belonging to one hemisphere were aligned, along the anterior–posterior dimension, using blood vessels that served as landmarks (blood vessels traversing across sections provide robust indication of the corresponding locations between two neighboring sections). The split images belonging to the other hemisphere were aligned similarly. The aligned images now occupying 3D volume were then aligned to rat brain atlas using posterior end of corpus callosum and dentate gyrus as reference. The background fluorescence (calculated from part of the section that contained no visible CTB-594 label) was subtracted for all images and the intensity of labeling across the entire cortical column was summed. The resulting summed intensity values in each hemisphere were normalized separately to the maximum of summed intensities in that hemisphere. The normalized summed intensity was then projected onto the top view and subsequently from top view images of all animals the mean image was calculated. Only the CTB-594 labeling in secondary visual mediolateral area (V2ML), primary visual cortex (V1), and secondary visual lateral area (V2L) was estimated. The width of the labeled band of neurons in medial–lateral dimension was estimated as the boundary in top view image along that dimension outside which intensity drops below 2% of maximum value (the width of the band was calculated at a distance of 2.4 mm anterior from interaural line). Correction of Ca²⁺-imaging data for in-frame-motion was performed as described in ref. [44] and analysis of neuronal Ca²⁺-transients and detection of putative action-potential-related Ca²⁺-signals was performed as described in ref. [26].

Multiphoton datasets from different animals were aligned on the basis of the intrinsic optical signal imaging response to visual stimulation (alignment procedure shown in Supplementary Fig. 1, intrinsic imaging described above). The center of mass of the 90th percentile contour determined from the intrinsic imaging response served as the global reference point, with datasets from different animals overlayed by alignment of this point. The 90th percentile contour determined from the intrinsic optical signal imaging response data (Supplementary Fig. 1a) was overlayed first on the image of the cortical vasculature (Supplementary Fig. 1b). The location of the center of mass was determined with reference to the surface vasculature pattern, and the corresponding point located manually in multiphoton images of the surface vasculature above the neurons from which functional multiphoton imaging data were acquired (Supplementary Fig. 1c). Finally, the location of the individual neurons in the multiphoton image of the surface vasculature were determined in images within layer 2/3 of z-stacks containing the images of the surface vasculature (Supplementary Fig. 1c, d).

Response probabilities in Fig. 3 and Supplementary Fig. 3a were determined during the initial 600 ms of the visual stimulus. For the neuronal population analysis in Fig. 3e, $N = 41$ neurons were used. To compare population spiking characteristics for Fig. 3f, g, from these 41 neurons, 5 neuronal populations with 6 neurons each were randomly selected.

To define responsive neurons, the spike rates resulting from presentations of orientated grating stimuli and a blank stimulus were subjected to Kruskal–Wallis nonparametric ANOVA followed by one-tailed Fisher's test. Monocular responsive neurons were defined as those whose evoked spike rate during presentation of at least one drift direction of orientated grating to an eye was significantly higher ($P < 0.05$) than evoked spike rates from other drift directions presented to the same eye or spontaneous spike rate. In addition to Kruskal–Wallis nonparametric ANOVA, also Fisher's test was used in cases where evoked spike rate of neurons was low (<1 spike per stimulus presentation) because in such cases the Kruskal–Wallis test would result in a spurious outcome. In brief, for calculating the contingency table used in Fisher's test, the peak orientation (defined below) and blank stimulus presentations that elicited one or more spikes were marked as being responsive and the presentations during which no spikes were elicited were marked as being non-responsive. The nos. of the presentations marked responsive and non-responsive for peak orientation and blank stimulus were then used in the contingency table for a one-tailed Fisher's test ($P < 0.05$). Binocularly responsive neurons were defined as those whose evoked spike rate during presentations to either eye of orientated grating of at least one drift direction was significantly higher than evoked spike rates from other drift directions presented to the respective eye or spontaneous spike rate, estimated using the same above measure as for monocular neurons.

Peak orientation was defined as that orientation of rectangular grating drifting in a direction that elicited maximum spike rate compared to rectangular drifting gratings of other orientations.

Preferred orientation was calculated from the phase angle of vector sum ($S$), as described in ref. [45], using the formula:

$$S = \sum_k R_k e^{i2\theta_k} \tag{1}$$

$$\text{Preferred orientation} = \frac{\arg(S)}{2} \tag{2}$$

$R_k$ represents mean spike rate for oriented grating drifting in direction $\theta_k$.

This method provides a robust estimate of preferred orientation, a continuous quantity, when sampling a neuron's orientation responses with a discrete series of grating orientations. The estimates obtained are robust to scenarios, such as the neuron's actual preferred orientation not falling on one of the grating orientations.

The orientation selectivity index (OSI) was calculated as described in ref. [46] as the magnitude of the normalized vector sum ($\hat{S}$) calculated using the formula:

$$\hat{S} = \frac{\sum_k R_k e^{i2\theta_k}}{\sum_k R_k} \tag{3}$$

Contralateral-eye bias index (CBI) was calculated as

$$\text{CBI} = \frac{\frac{\sum_k R_k^{\text{Contra–eye}}}{N}}{\frac{\sum_k R_k^{\text{Contra–eye}}}{N} + \frac{\sum_k R_k^{\text{Ipsi–eye}}}{N}} \tag{4}$$

$\sum_k \frac{R_k^{\text{Contra–eye}}}{N}$ & $\sum_k \frac{R_k^{\text{Ipsi–eye}}}{N}$ denotes mean stimulus evoked spike rate during contralateral and ipsilateral eye stimulations, respectively. $N$ represents number of drifting directions.

Circular statistics was performed using CircStat toolbox[47].

The vc reduction index was calculated by first adjusting the spike rate of neurons whose responses were abolished during vc inactivation to zero. The vc reduction index was then calculated as

$$\text{vc reduction index} = \frac{\text{Spike rate for peak angle during LEDoff} - \text{Spike rate for peak angle during LEDon}}{\text{Spike rate for peak angle during LEDoff}} \tag{5}$$

with any values from the above computation that were negative being rounded to zero. This then results in values ranging from zero to one, with zero representing activity that is completely unreliant on the vc pathway and one representing activity that is totally reliant on the vc pathway. For monocular neurons, this value was calculated only for the responses from the active eye. For binocular neurons, the index was calculated separately for responses from each eye and combined vc reduction index calculated as the mean of these. Any negative values were rounded to zero.

**Analysis of neuropil-related Ca²⁺-fluorescence signal**. A region of interest (ROI) excluding the neuronal somatas and astrocytes, and about 1/5 the total size of the image was marked to obtain the neuropil-related Ca²⁺-fluorescence signal. To assess the neuropil signal driven from stimulation of either contralateral or ipsilateral eye simultaneously with contralateral non-silencing/silencing, the signal associated with all periods of rectangular grating display (all drift directions and all repetitions) for each eye and non-silencing/silencing condition was combined, and compared with the neuropil signal derived from periods of same duration as stimulus but whose occurrence was randomly chosen. As the stimulation periods occupied a large portion of the imaging duration (42.9%), the procedure of randomly choosing durations was modified so as to assure that shuffling through different random durations resulted in at least 50% of the chosen durations not coinciding with stimulation. After the induction of thalamic silencing (MUSC), the average neuropil signal driven from contralateral eye stimulation and simultaneous contralateral non-silencing (LED off)/silencing (LED on) was significantly higher than the average signal associated with randomly chosen durations (Rat 1; MUSC + LED off: 3.27 ± 0.25%ΔF/F, mean ± SEM, 1-tailed 2-sample $t$-test, $P < 0.001$, MUSC + LED on: 2.87 ± 0.26%ΔF/F, mean ± SEM, 1-tailed 2-sample $t$-test, $P = 0.004$; Rat 2; MUSC + LED off: 2.8 ± 0.19%ΔF/F, mean ± SEM, 2-sample $t$-test, $P = 0.004$, MUSC + LED on: 2.82 ± 0.2%ΔF/F, mean ± SEM, 2-sample $t$-test, $P = 0.003$). Whereas for ipsilateral eye stimulation, the average signal with simultaneous contralateral non-silencing was significantly higher than the average signal associated with randomly chosen durations, the average signal with simultaneous contralateral silencing did not differ from the average signal associated with randomly chosen durations (Rat 1; MUSC + LED off: 2.63 ± 0.22%ΔF/F, mean ± SEM, 1-tailed 2-sample $t$-test, $P = 0.027$, MUSC + LED on: 1.95 ± 0.17%ΔF/F, mean ± SEM, 1-tailed 2-sample $t$-test, $P = 0.854$; Rat 2; MUSC + LED off: 2.68 ± 0.19% ΔF/F, mean ± SEM, 1-tailed 2-sample $t$-test, $P = 0.022$, MUSC + LED on: 2.48 ± 0.2%ΔF/F, mean ± SEM, 1-tailed 2-sample $t$-test, $P = 0.185$).

**Statistics and reproducibility**. The multiphoton overview images presented in Fig. 1e, h are representative of 22 fields of view acquired from 17 animals with similar results. The micrographs and multiphoton overview image presented in Fig. 2a are representative of five fields of view from three animals. The micrographs shown in Fig. 3a are representative of two experiments with similar results. The multiphoton overview in Fig. 5e is representative of two fields of view from two animals. The data presented in Supplementary Fig. 1 is representative of 17

experiments with similar results. The micrographs and multiphoton overview images presented in Supplementary Fig. 3 are representative of three experiments with similar results. The micrograph presented in Supplementary Fig. 4a is representative of 10 experiments with similar results.

**Reporting summary**. Further information on research design is available in the Nature Research Reporting Summary linked to this article.

## Data availability
Data available on request.

## Code availability
Code supporting the analyses and figures presented in the study available from the authors on reasonable request.

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

## Acknowledgements
We thank Heinz Beck and Richard Hahnloser for comments on an earlier version of this manuscript. We thank U. Czubayko for technical help with histology. We thank Michael Straussfeld and Rolf Honnef from the mechanical workshop. We thank Juan Daniel Flórez Weidinger and Fred Wolf for valuable insights and help with designing initial experiments. We thank the Max Planck Society for support as well as Stiftung caesar.

## Author contributions
Study conceived and designed by V.R., V.P., J.N.D.K. Data collection V.R. and V.P. Analysis V.R., V.P., D.J.W., and J.N.D.K. Initial draft written by V.R. and J.N.D.K. This version of the manuscript written by V.P., D.J.W., and J.N.D.K.

## Competing interests
The authors declare no competing interests.
