## [Peer Review File · Nature Communications]

Reviewers' comments:

Reviewer #1 (Remarks to the Author):

This manuscript uses state of the art techniques to address the question of the role of inputs from the corpus callosum to the visual cortex in the rodent, the rat in this case. This question has been addressed for decades in experiments that failed to answer it definitively. Indeed, before reading this manuscript, I had believed that the major function of callosal input to V1 was to convey strong ipsilateral-eye input to the binocular zone of the other hemisphere in order to account for the much more nearly equal responses to the two eye than seemed possible on the basis of input from the lateral geniculate nucleus alone. I had believed this even though it had been contradicted by some papers in the literature that I assessed to be weak and supported by others, that were, unfortunately, similarly weak. So it is a real gift to read a paper that I find fully convincing on its main points, which employs adequate methods that have become available only in the past few years, and which collects a sufficient body of data.

One view of the callosum is that it merely continues the lateral interactions that are found within each hemisphere across the place where the hemispheres are divided. This view is attractive, but it neglects the fact of some degree of overlap in the visual field representations in the two hemispheres. I think that this issue does, however, merit at least brief mention, although a full discussion of it would require a different experiment that would be far beyond the reasonable scope of this manuscript.

The manuscript could be more concise, although the wealth of statistical comparison, while appropriate, makes it hard to read many of the sentences. I hope that journals will change their style to make manuscripts easier to read by moving the statistical comparisons that are now set out in the text, interrupting nearly every sentence in the results, to a list at the end of Methods, referred to in the results by a tiny superscript.

The experiments are fully compelling with one exception noted below. I have only a few minor comments.

1. The phrase at line 36 and elsewhere about the ipsilateral eye's "uncrossed projection to the contralateral visual cortex" is VERY hard to understand because "projection" is normally used to indicate an anatomical connection, and the authors use it incorrectly, in my view, to refer to a 3-step synaptic relay. Maybe arrowheads in Figure 1d to indicate the alternative pathway [ipsi eye to contra LGN to contra V1 then via callosum to ipsi V1) would help, and labels to indicate sites of recording and optogenetic perturbation
2. So much else is set out in detail, but Fig 1 panel I is not clear. To make panel i clear, you need to show at least one intrinsic signal imaging example, and better a pair of them.
3. In Figure 3d gray is a terrible color , almost invisible. Try blue or something else.
4. In lines 382-382 the authors use the phrase "spatially non-uniform" in reference to the comparison between the callosal-projecting and non-projecting cells. This is needlessly confusing phraseology, and I would see this as uniform—the VCPN and non-VCPN cells are identical. This is a question of writing, not of fact. Please revise.
5. The more serious question, and the only potential artefact that I see in the results, is that this comparison between VCPN and non-VCPN cells rests on the assumption that ALL of the callosal-projecting cells have been labeled. This seems unlikely, and the authors need to address this issue.
6. While I am generally reluctant to object to discussion points raised by the authors so long as they

do not claim to have demonstrated more than their findings actually demonstrate, I still find the speculation of function in lines 456-470 quite weak, and I think that omitting it would improve the manuscript if they can not strengthen it.

Michael Stryker

Reviewer #2 (Remarks to the Author):

The manuscript by Ramachandra and others package a series of experiments employing optogenetic and pharmacological manipulation combined with two-photon imaging and retrograde anatomical tracing. These experiments are well structured to demonstrate that the visual callosal pathway works in concert with ipsilateral LGN to generate visually evoked suprathreshold activity in V1. Importantly, the paper was able to demonstrate that the callosal pathway alone is not sufficient to generate the suprathreshold activity in V1. These findings are novel and will have a good chance of being of interest to the community of visual neuroscience. Moreover, these findings may impact the neuroscience community more broadly, as it significantly contributed to the elucidation of the functionality of the callosal pathway, which remains mostly mysterious to date. While this is the case, I do have some suggestions for further clarification as follows.

(I.112) Where in the visual field were the drifting grating stimuli presented?

(I.147) Details of the intrinsic optical imaging are missing. How was the imaging data analyzed to determine a small red spot in Fig3a?

(I.291, I.293) Please specify which panel amongst the three panels in Fig4f does the sentence refer to.

(I.316) The word "sub-region" is misleading as it implies a spatially confined domain of the primary visual cortex. Such a confined domain is not explicitly shown in the current manuscript. It would be interesting to investigate if there exists a confined region to which the vc pathway projects more effectively (as shown in the previous anatomical study by Laing et al., 2015). One way to study this would be to visualize locations of each neuron in V1 (as in the format similar to Figure 1i), where each neuron is color-coded by the "VC reduction index" employed in Figure 4f.

(I.328-329, "confirm that vs inactivation had the same effect as described before (compare Fig. 5d and 4a")) A reader will find comparing Fig5d with 4a difficult, as these figures are presented in different formats. Data is divided by the three cell classes, whereas in Fig4a, and by stimulus type in Fig5a.

(I.332-333, "a subpopulation of neurons were observed whose responses to stimulation of the ipsilateral eye were abolished on vc pathway silencing (Fig. 5d, N = 2 neurons)") Please specify the two neurons in Fig5d.

(II.397-398) A reader may find it informative if the discussion includes the following topic: how much of the result is specific to rodents?

(I.413-415 "we found no significant influence of inactivation on orientation tuning ... suggests that the axons of neurons projecting through the vc pathway target contralateral neurons with similar orientation preference") It is not intuitively clear why the finding leads to this suggestion. Given the result of the inactivation experiment, it also seems possible that the axons of neurons projecting through the vc pathway target contralateral neurons without orientation preference.

(I.660) Please add information on the effective power density of the LED on the cortical surface.

(I.661) Please add a specification of the lens, such as its shape and manufacturer.

(I.720-721) Please add a description of how the automated shutters enabled monocular stimulation.

(Fig. 2a) The left bottom image of the panel (layer dependent expression) shows some level of fluorescence expression in L1, giving an impression not consistent with panel b. A reader may wonder where the source of the fluorescence stems from. A magnified image of the L1 (, similar to the right bottom image of panel a) would be informative.

(Fig. 4b, c) Please superimpose the traces from the top and bottom rows, so the effect of vc inactivation is readily visible.

(Fig. 4e) The panel serves to convey one of the main results, yet it isn't very easy to understand. To facilitate understanding of readers, it is advisable to add another example cell, which shows a significant reduction to the contra stimulation and vc inactivation (open green symbol) in the format similar to panels b and c. What is the advantage of using "delta" as opposed to the "VC reduction index" employed in panel g?

Reply to reviewers' comments:

Reviewer #1 (Remarks to the Author):

This manuscript uses state of the art techniques to address the question of the role of inputs from the corpus callosum to the visual cortex in the rodent, the rat in this case. This question has been addressed for decades in experiments that failed to answer it definitively. Indeed, before reading this manuscript, I had believed that the major function of callosal input to V1 was to convey strong ipsilateral-eye input to the binocular zone of the other hemisphere in order to account for the much more nearly equal responses to the two eye than seemed possible on the basis of input from the lateral geniculate nucleus alone. I had believed this even though it had been contradicted by some papers in the literature that I assessed to be weak and supported by others, that were, unfortunately, similarly weak. So it is a real gift to read a paper that I find fully convincing on its main points, which employs adequate methods that have become available only in the past few years, and which collects a sufficient body of data.

One view of the callosum is that it merely continues the lateral interactions that are found within each hemisphere across the place where the hemispheres are divided. This view is attractive, but it neglects the fact of some degree of overlap in the visual field representations in the two hemispheres. I think that this issue does, however, merit at least brief mention, although a full discussion of it would require a different experiment that would be far beyond the reasonable scope of this manuscript.

We acknowledge this point and agree that mentioning this is of relevance for this manuscript, and have added the following text the introduction (p. 3, new text in italics):

"...However, differences in rodent visual system, compared to cat and primate, at the level of the projections from the retinae^{14,15} and the central visual centers^{16,17}, hinder clear comparisons of callosal function. One suggestion for the function of the callosal projection is that it facilitates lateral interactions between neurons representing adjacent regions in the visual field across the location where the visual field is divided between the hemispheres^{8,18,19}. This has been proposed for primates and cats where the visual field is divided along the vertical meridian with the left and right halves of the visual field (hemifields) being represented in the right and left hemispheres respectively. However, in many other animals including rodents^{8,16,20,21}, there is considerably more overlap in the representation of the visual field around the vertical meridian in the two cortical hemispheres, and lateral connectivity can in principle be achieved without the callosal projection..."

The manuscript could be more concise, although the wealth of statistical comparison, while appropriate, makes it hard to read many of the sentences. I hope that journals will change their style to make manuscripts easier to read by moving the statistical comparisons that are now set out in the text,

interrupting nearly every sentence in the results, to a list at the end of Methods, referred to in the results by a tiny superscript.

We agree that there are many occurrences in the text where the flow of a sentence is interrupted by statistics in parentheses, and in retrospect in some cases the flow could be improved by grouping statistics better, and moving statistics to the end of a sentence. We have made modifications to the text in numerous locations to try to improve sentence flow along these lines. Some examples are given below.

p5. Now: "...In this labelled region we quantified L2/3 neurons' suprathreshold response properties to monocular and binocular visual stimulation, using Ca^{2+} -transients (calcium-sensitive fluorescence indicator Oregon green-BAPTA 1 (OGB1), Fig. 1e-f, see Fig. 1c,d for experimental setup and terminology)."

Previous: "...In this labelled region we quantified L2/3 neurons' suprathreshold response properties to monocular and binocular visual stimulation (see Fig. 1c,d for experimental setup and terminology), using Ca^{2+} -transients (calcium-sensitive fluorescence indicator Oregon green-BAPTA 1 (OGB1), Fig. 1e-f)."

p9. Now: "...While both monocular contralateral and binocular stimuli elicited similar probability distributions, ipsilateral stimulation generated significantly less spiking events and more trials in which no spikes were evoked, which was also significantly more than observed during spontaneous activity (*binocular vs contralateral*, Kolmogorov-Smirnov test $P = 0.299$; *binocular vs ipsilateral*, Kolmogorov-Smirnov test $P < 0.0001$; *ipsilateral vs spontaneous*; Kolmogorov-Smirnov test $P < 0.0001$, Fig. 3f, $N = 30$ neurons, $N = 5$ populations with 6 neurons each)."

Previous: "...While both monocular contralateral and binocular stimuli elicited similar probability distributions (*binocular vs contralateral*; Kolmogorov-Smirnov test $P = 0.299$), ipsilateral stimulation generated significantly less spiking events and more trials in which no spikes were evoked (*binocular vs ipsilateral*; Kolmogorov-Smirnov test $P < 0.0001$), which was also significantly more than observed during spontaneous activity (*ipsilateral vs spontaneous*; Kolmogorov-Smirnov test $P < 0.0001$, Fig. 3f, $N = 30$ neurons, $N = 5$ populations with 6 neurons each)."

p9. Now: "...As with the spiking statistics, both binocular and contralateral stimulation evoked similar probability distributions of active neurons per stimulus, whereas ipsilateral stimulation evoked significantly less active neurons (Fig. 3g; *binocular vs contralateral*, Kolmogorov-Smirnov test $P = 0.8532$, excluding 0 bin; *binocular vs ipsilateral*; Kolmogorov-Smirnov test $P < 0.0001$, excluding 0 bin)."

Previous: “...As with the spiking statistics, both binocular and contralateral stimulation evoked similar probability distributions of active neurons per stimulus (excluding 0 bin; binocular vs contralateral; Kolmogorov-Smirnov test $P = 0.8532$), whereas ipsilateral stimulation evoked significantly less active neurons (Fig. 3g, excluding 0 bin; binocular vs ipsilateral; Kolmogorov-Smirnov test $P < 0.0001$).”

p11. Now: “...This analysis showed that the ipsilateral crossed pathway (ipsi-eye -> contra-LGN -> contra-V1 -> ipsi-V1 via vc) reduced most in binocular neurons, especially in neurons where responses after vc pathway inactivation were indistinguishable from spontaneous rates (Figure 4c, example and Figure 4e, open symbols, Supplementary Figure 11a, ipsi- vs contralateral stimulus response reduction, $54.08 \pm 26.88\%$ vs $35.95 \pm 29.51\%$, mean \pm SD, Mann-Whitney U test $P = 0.036$).”

Previous: “...This analysis showed that the ipsilateral crossed pathway reduced most in binocular neurons (Supplementary Fig. 7, ipsi- vs contralateral stimulus response reduction, $54.08 \pm 26.88\%$ vs $35.95 \pm 29.51\%$, mean \pm SD, Mann-Whitney U test $P = 0.036$), especially in neurons where responses after vc pathway inactivation were indistinguishable from spontaneous rates (Fig. 4c, example and Fig. 4e, open symbols).”

The experiments are fully compelling with one exception noted below. I have only a few minor comments.

1. The phrase at line 36 and elsewhere about the ipsilateral eye's "uncrossed projection to the contralateral visual cortex" is VERY hard to understand because "projection" is normally used to indicate an anatomical connection, and the authors use it incorrectly, in my view, to refer to a 3-step synaptic relay. Maybe arrowheads in Figure 1d to indicate the alternative pathway [ipsi eye to contra LGN to contra V1 then via callosum to ipsi V1] would help, and labels to indicate sites of recording and optogenetic perturbation

Michael Stryker is absolutely correct to draw attention to this sentence, which is in fact a typographical error on our part. We have corrected this as follows (p. 3, new text in italics):

“...uncrossed projection to the ipsilateral LGN (see Fig. 1c for schematic). An alternative anatomical pathway giving rise to responses from the ipsilateral eye results from *its tri-synaptic projection involving the contralateral visual cortex and projection through the corpus callosum (ipsilateral eye – contralateral LGN – contralateral visual cortex – ipsilateral visual cortex via the callosal projection)*³⁻⁵.”

In addition, we acknowledge that the nomenclature at other points in the text is potentially confusing. To address this, we have, at the expense of additional sentence length, added specific descriptions of the pathway being referred to. In the case of the phrase on line 36 referred to

above, we have added this description in parentheses at the end of the sentence (quoted above). In addition, we have added on p. 11 (new text in italics):

“...This analysis showed that the ipsilateral crossed pathway (*ipsi-eye -> contra-LGN -> contra-V1 -> ipsi-V1 via vc*) reduced most in binocular neurons...”

And on p 17 (new text in italics):

“...The finding that the neuropil signal remaining during VCPN inactivation is abolished by inactivation of the ipsilateral LGN (Fig. 5) is consistent with there being only two possible pathways from ipsilateral eye to visual cortex, the uncrossed projection from the ipsilateral retina and the crossed projection to the contralateral LGN and visual cortex via the callosal projection (*ipsilateral eye – contralateral LGN – contralateral visual cortex – ipsilateral visual cortex via corpus callosum*)....”

As suggested, we have also added arrows indicating the two potential anatomical pathways from the ipsilateral eye into Fig. 1d with accompanying description in the figure legend.

2. So much else is set out in detail, but Fig 1 panel I is not clear. To make panel i clear, you need to show at least one intrinsic signal imaging example, and better a pair of them.

It was an omission on our part not to have included a more comprehensive description of the method used for aligning the multiphoton data using the intrinsic optical signal imaging response. We have added to the revised manuscript a new paragraph in the Methods section in which this is described, as well as a new Supplementary Figure (Supplementary Figure 1) illustrating the method. The new text reads as follows (p.24):

“Multiphoton datasets from different animals were aligned on the basis of the intrinsic optical signal imaging response to visual stimulation (alignment procedure shown in Supplementary Figure 1, intrinsic imaging described above). The center of mass of the 90th percentile contour determined from the intrinsic imaging response served as the global reference point, with datasets from different animals overlaid by alignment of this point. The 90th percentile contour determined from the intrinsic optical signal imaging response data (Supplementary Figure 1a) was overlaid first on the image of the cortical vasculature (Supplementary Figure 1b). The location of the center of mass was determined with reference to the surface vasculature pattern, and the corresponding point located manually in multiphoton images of the surface vasculature above the neurons from which functional multiphoton imaging data were acquired (Supplementary Figure 1c). Finally, the location of the individual neurons in the multiphoton image of the surface vasculature were determined in images within layer 2/3 of z-stacks containing the images of the surface vasculature (Supplementary Figure 1c,d).”

The new Supplementary Figure is referred to in the text on p. 5 as follows:

“...Based on intrinsic optical signal imaging response (see methods and Supplementary Figure 1), all fields of view were overlaid showing that both monocular and binocular neurons were interspersed and showed no obvious spatial clustering...”

3. In Figure 3d gray is a terrible color, almost invisible. Try blue or something else.

We have modified the figure as suggested. The same change has also been made for the schematics in Fig. 1c and 5a.

4. In lines 382-382 the authors use the phrase “spatially non-uniform” in reference to the comparison between the callosal-projecting and non-projecting cells. This is needlessly confusing phraseology, and I would see this as uniform—the VCPN and non-VCPN cells are identical. This is a question of writing, not of fact. Please revise.

The summary description of the spatial organization of V1 neurons has been modified in the revised text to remove this phrase, and now reads as follows (p. 15, new text in italics):

“...previously reported^{11,13,18-20}. We show that neuronal populations in lateral binocular V1 contain a mixture of projecting, i.e. VCPNs, and non-projecting neurons, non-VCPNs. These populations do not show obvious spatial organization, with VCPNs being often direct neighbors to non-VCPNs.”

5. The more serious question, and the only potential artefact that I see in the results, is that this comparison between VCPN and non-VCPN cells rests on the assumption that ALL of the callosal-projecting cells have been labeled. This seems unlikely, and the authors need to address this issue.

The ideal case (in the absence of having a method capable of labelling or infecting 100% of the callosally projecting neurons) would be to be able to quantify the absolute number of neurons in V1 projecting through the corpus callosum. However, we do not currently see a way to provide this quantification experimentally. We have addressed this limitation in an additional paragraph in the discussion specifically about this topic. This paragraph reads as follows (p.15):

“Any study using retrograde tracing or virus-mediated expression techniques is limited at least to some extent by the total fraction of the target neuronal population labelled by the technique.

In the current study, this applies both to the combined retrograde tracing and Ca²⁺-imaging experiments as well as to the optogenetic manipulation of V1 contralateral to the recorded V1 neurons. For the retrograde tracing experiments, we found that around 35% of neurons in layer 2/3 were labelled by the labelling technique. Compared to the unlabeled neurons, the labelled population of projecting neurons did not show significant differences in tuning and response properties to the visual stimuli we presented, though the possibility still remains that any unlabeled projection neurons may have more selective visual response properties. A similar possibility exists also for the optogenetic manipulation experiments. However, given that an appreciable fraction of neurons show significant modulation of responses, including reduction of evoked responses to the level of spontaneous activity, we clearly show that activity in callosally projecting neurons is an important component of visual responses for many V1 neurons.”

6. While I am generally reluctant to object to discussion points raised by the authors so long as they do not claim to have demonstrated more than their findings actually demonstrate, I still find the speculation of function in lines 456-470 quite weak, and I think that omitting it would improve the manuscript if they can not strengthen it.

We acknowledge that this paragraph was speculative, and have removed it as suggested.

Michael Stryker

Reviewer #2 (Remarks to the Author):

The manuscript by Ramachandra and others package a series of experiments employing optogenetic and pharmacological manipulation combined with two-photon imaging and retrograde anatomical tracing. These experiments are well structured to demonstrate that the visual callosal pathway works in concert with ipsilateral LGN to generate visually evoked suprathreshold activity in V1. Importantly, the paper was able to demonstrate that the callosal pathway alone is not sufficient to generate the suprathreshold activity in V1. These findings are novel and will have a good chance of being of interest to the community of visual neuroscience. Moreover, these findings may impact the neuroscience community more broadly, as it significantly contributed to the elucidation of the functionality of the callosal pathway, which remains mostly mysterious to date. While this is the case, I do have some suggestions for further clarification as follows.

(I.112) Where in the visual field were the drifting grating stimuli presented?

While the location of the visual stimulus presentation was described in the Methods section (subsection *Visual Stimulation*), it was remiss of us not to provide more detail in the results section. We have now added a description of the stimulus location and reference to the Methods section at this line in the revised text as follows (p. 6, new text in italics):

“The mean spiking rate of the VCPN subpopulation evoked by drifting grating stimuli was not significantly different to that of non-VCPN subpopulation (VCPN vs non-VCPN, 0.32 ± 0.22 Hz vs 0.4 ± 0.29 Hz, mean \pm SD, Mann-Whitney U test $P = 0.437$, $N = 28$ (VCPN) & $N = 33$ (non-VCPN), *gratings presented in the visual space in front of the nose, see Methods for details*).”

(I.147) Details of the intrinsic optical imaging are missing. How was the imaging data analyzed to determine a small red spot in Fig3a?

The small red spot in Fig. 3a is a small injection of Alexa-conjugated cholera toxin B (CTB) made into layer 2/3 in the center of the region identified using intrinsic optical signal imaging as representing the visual space in front of the animal. This CTB injection can then be directly visualized in the post hoc histology. Full details of the analysis of the intrinsic imaging data are given in the Methods section (subsection *Intrinsic imaging of visually related brain activity*) of the submitted manuscript as follows (l. 688-691):

“The images acquired before the onset of stimulus presentation were averaged together, with the result used for background subtraction. Image frames between stimulus onset and offset were averaged and a contour at the 90 percentile used as indicative of the region activated by the visual stimulus.”

In retrospect, the description of this experiment in the results section of the submitted manuscript was not clear, and we have expanded the description in the revised text to further clarify how this experiment was performed. The revised text reads as follows (p. 7, new text in italics):

“...border region and spanning all cortical layers (Fig. 3a). We next identified cortical representation of the binocular visual space directly in front of the animal using intrinsic optical signal imaging, and made a small injection of Alexa-conjugated CTB into layer 2/3 in the center of the identified region (small red spot in Fig. 3a, see methods for full details). This showed that the V1-V2 border region receiving the afferent callosal projection overlaps with the representation of the frontal binocular visual field.”

(I.291, I.293) Please specify which panel amongst the three panels in Fig4f does the sentence refer to.

The description at the beginning of the paragraph including I. 291 and 293 in the original text refers to the right panel in Fig. 4f, with the first sentence considering the outcomes for stimulation of the ipsilateral eye, and the second the outcomes for stimulation of the contralateral eye. Both references to Fig. 4f have been amended to include a reference to the right panel as follows (p.12, new text in italics):

“For a binocular neuron, a reduction in responsiveness upon stimulation of the ipsilateral eye by inactivating the vc pathway implies that the pathway must involve the crossed projection through the contralateral LGN to contralateral V1, and then the vc pathway from contralateral V1 (Fig. 4f, right panel). It follows that a reduction in responsiveness upon stimulation of the contralateral eye implies that the pathway must involve the uncrossed projection from the contralateral eye through the contralateral LGN to contralateral V1 and then the vc pathway from contralateral V1 (Fig. 4f, right panel).”

(I.316) The word "sub-region" is misleading as it implies a spatially confined domain of the primary visual cortex. Such a confined domain is not explicitly shown in the current manuscript. It would be interesting to investigate if there exists a confined region to which the vc pathway projects more effectively (as shown in the previous anatomical study by Laing et al., 2015). One way to study this would be to visualize locations of each neuron in V1 (as in the format similar to Figure 1i), where each neuron is color-coded by the "VC reduction index" employed in Figure 4f.

With regard to the use of “sub-region” in the sentence in question, we agree with the reviewer that this is misleading, and have revised the sentence to now read (p.12-13, new text in italics):

“...The above analyses suggest that the vc pathway plays a crucial role in generating spiking responses in within its innervation domain in binocular primary visual cortex, but this does not

necessarily imply that there are neurons wholly reliant on this pathway for their inputs or subthreshold activity. While our approach..."

With regard to potential clustering of neurons whose responses are modulated by vc pathway inactivation, we did not observe any such clustering in the analysis suggested by the reviewer. These results have been included on p.12 and in Supplementary Fig. 11b, and reads as follows (new text in italics):

"...population of monocular neurons in V1. In addition, although callosally projecting axons have been observed to accumulate into patches along the lateral edge of V1 in flattened-cortex histological preparations from rats¹¹, we did not observe any clustering of neurons whose responses were more strongly modulated by vc pathway inactivation in an overlay map of all recorded neurons color-coded by their vc reduction index (Supplementary Fig. 11b)."

(I.328-329, "confirm that vs inactivation had the same effect as described before (compare Fig. 5d and 4a)") A reader will find comparing Fig5d with 4a difficult, as these figures are presented in different formats. Data is divided by the three cell classes, whereas in Fig4a, and by stimulus type in Fig5a.

While we can see the reviewers point here, our objective in presenting the data the way we did in Fig. 4a and Fig. 5d was to present as much of the data as possible. As we wanted to present all data in Fig. 4a, we decided to separate the cells out by ocular dominance, as then we could also show all data from cells whose responses were not influenced by vc pathway inactivation. In Fig. 5d, we have the additional groups from the muscimol control and muscimol with inactivation, with this being the critical part of the data being presented. Presenting the data as ipsilateral stimulation and contralateral stimulation makes the distinction between monocular and binocular neurons obscure, we agree, but the key point is that there the same range response modulation by vc pathway inactivation was observed. We are still of the opinion that the way the data is presented in Figs. 4a and 5d are the most concise and appropriate means for displaying this data and have respectfully not made alterations in response to this comment.

(I.332-333, "a subpopulation of neurons were observed whose responses to stimulation of the ipsilateral eye were abolished on vc pathway silencing (Fig. 5d, N = 2 neurons)") Please specify the two neurons in Fig5d.

The two ipsilateral responses that were abolished on vc pathway silencing are shown connected by orange lines in the revised version of the figure and referred to as such in the accompanying figure legend.

(II.397-398) A reader may find it informative if the discussion includes the following topic: how much of the result is specific to rodents?

We have added a new paragraph to the discussion addressing this topic. It reads as follows (p. 16):

“The callosal projection in the visual cortex has been observed in many mammalian species including cat^{4,5,33}, rat⁶, mouse^{6,34}, ferret⁷, tree shrew^{8,35}, rabbit⁹ and monkey¹⁰, with projecting neurons being present in regions of visually receptive cortex representing the area of visual space around the vertical meridian. Considering the function of the callosal pathway initial studies in cats showed that the projection targets the retinotopically matched region of contralateral visual cortex³ and that inhibition of CPNs by cortical cooling has a variety of consequences for the visual responses of neurons in the recipient cortical region including a general reduction in activity, loss of the receptive field from one eye or loss of a portion of the RF from both eyes³⁶. It has also been suggested that the callosal projection plays the role of unifying cortical regions with matching location in the visual field and/or visual response properties^{8,37}. In tree shrews, the callosal projection targets regions with visuotopic correspondence, but without preferentially projecting to areas of similar orientation selectivity⁸. In the cat, however, there is also both anatomical and physiological data suggesting that the callosal projection not only targets retinotopically matched regions but also regions with similar orientation preference³⁷⁻³⁹. Moreover, the effects of contralateral cooling have been found to result in a preferential modulation of responses from neurons with vertical and horizontal orientation preference³⁹. The findings in the current study on the orientation tuning of neurons in the rat visual cortex indicate that the callosal projection in this species does not appear to selectively target neurons with a particular orientation preference, and leaves orientation tuning unchanged. Further, for the cat, inhibition of the callosal projection in rats has a variety of consequences for spiking activity in the recipient neuronal populations, ranging from no effect to complete elimination of visual responses. One key finding in the current study is the callosal projection in rats is only able to drive spiking responses in their projection targets when there is simultaneous activity in the pathway from the ipsilateral LGN. Given that there are broad similarities in callosal anatomy and function across the different species described above it seems possible that this might also be the case for the callosal projection in other species too.”

(l.413-415 "we found no significant influence of inactivation on orientation tuning ... suggests that the axons of neurons projecting through the vc pathway target contralateral neurons with similar orientation preference") It is not intuitively clear why the finding leads to this suggestion. Given the result of the inactivation experiment, it also seems possible that the axons of neurons projecting through the vc pathway target contralateral neurons without orientation preference.

We thank the reviewer for drawing attention to this point and acknowledge their concern about this statement. In retrospect, this idea had been condensed in the text to the point where the supporting data was not at all clearly described or presented. The suggestion that there may be some selective targeting of callosal axons to neurons with similar orientation selectivity comes from the following findings in the current dataset. First, strongly orientation tuned responses recorded from some neurons were abolished (reduced to spontaneous rates) during vc pathway

inactivation (for example the ipsilateral eye response shown in the left column, third row in Supplementary figure 5 of the submitted manuscript, now Supplementary Figure 7). This would suggest that for the ipsilateral eye response for this neuron, all orientation tuning is due to the orientation tuning of the callosally-projecting neuronal population connecting with it (whether they be mono- or polysynaptic connections). Second, for strongly orientation tuned responses in other neurons (ie with stimulus-evoked responses only present for a restricted set of stimulus orientations), the amplitude of the response was reduced, but the range of orientation which evoked responses was not changed. One explanation for this would be that the callosally projecting neuron(s) had similar orientation tuning to the neuron under investigation.

We have expanded this section with the intention of making the background for the suggestion clearer. The section now reads as follows (p.17, new text in italics):

“In the current study we found no significant influence of inactivation of the vc pathway on orientation-tuning properties of V1 neurons. For some neurons, strongly orientation tuned responses were no longer present in the presence of vc pathway inactivation (see Supplementary Fig. 5 & 6), suggesting that the orientation tuning was due to the orientation tuning of the pre-synaptic callosally projecting neurons. For other neurons, strongly tuned responses were reduced in response amplitude, but without changing the preferred orientation (see Supplementary Fig. 5 & 6). These findings raise the possibility that the axons of neurons projecting through the vc pathway target contralateral neurons with similar orientation preference to that of the projecting neurons, consistent with the findings of a recent study showing that...”

(I.660) Please add information on the effective power density of the LED on the cortical surface.

(I.661) Please add a specification of the lens, such as its shape and manufacturer.

We thank the reviewer for point out these inadvertent omissions. The effective power density at the surface was 12.8 mW/mm^2 , and the lens was a half-ball lens (S-LAH79, 2 mm diameter, Edmund Optics, NJ, USA). Both details have been added to the revised manuscript, and this section now reads (p. 20, new text in italics):

“...opened over the previous AAV-eArchT injection site and a LED (Golden Dragon, 590nm, OSRAM, Regensburg, Germany) with attached half-ball lens (S-LAH79, 2 mm diameter, Edmund Optics, NJ, USA) was mounted over the cortical site, with shielding to prevent the light from escaping the cortex. The effective power density of the LED at the cortical surface was 12.8 mW/mm^2 . For electrophysiology...”

(I.720-721) Please add a description of how the automated shutters enabled monocular stimulation.

We have added a supplementary figure (Supplementary Fig. 13) showing the shutter design and a description of the shutter design and function in the Methods section of the revised text, which reads as follows (p.22-23, new text in italics):

“...monocularly and in a subset of 5 animals also 8 times to both eyes (binocularly). The automated shutters were custom made, and consisted of a neoprene cup to cover the eye connected via a ~8 cm long rod to a servo motor (Supplementary Fig. 13). The neoprene cup was placed over the eye, pressing slightly into the surrounding fur, and could be retracted at an angle downwards away from the eye using the servo motor to allow visual stimulus presentation. The servo motor position was controlled via the visual stimulus software to allow full computer control and randomization of monocular or binocular stimulus presentation. Inter-stimulus interval was between...”

(Fig. 2a) The left bottom image of the panel (layer dependent expression) shows some level of fluorescence expression in L1, giving an impression not consistent with panel b. A reader may wonder where the source of the fluorescence stems from. A magnified image of the L1 (, similar to the right bottom image of panel a) would be informative.

While there is some diffuse red fluorescence all around the region of retrogradely labelled neurons, including in layer 1, neuronal somata containing cholera-toxin labelling were not observed. We have included an additional supplementary figure (Supplementary Figure 3) showing a further enlargement of the superficial region of a post hoc histological section from an animal in which callosally projecting neurons had been retrogradely labelled with Alexa-conjugated cholera toxin B. We have also included in this supplementary figure an image similar to that presented in Fig. 2a lower right but taken from layer 1. Labelling in neuronal somata is not apparent.

We refer to this Supplementary figure in the revised text as follows (p.6):

“...Starting from the surface, the first co-labelled neurons were located approximately 140 μ m below the pia, supporting the notion that no layer 1 neurons project to contralateral cortex³¹ (Figure 2b, Supplementary Figure 3).”

(Fig. 4b, c) Please superimpose the traces from the top and bottom rows, so the effect of vc inactivation is readily visible.

In preparing the response to this reviewer comment it came to our attention that we had made an error with our reference to this figure panel in the text. In the original text we stated:

“From this analysis it was clear that vc pathway inactivation had a wide range of effects on stimulus-evoked spiking in subpopulations of binocular and monocular neurons, which ranged from ineffective (Fig. 4a,b) to rendering neurons unresponsive to visual stimuli (Fig. 4c).”

In all analyses performed, the designation of unresponsiveness to visual stimulation was always made on the grounds of the spiking activity inferred from the recorded Ca^{2+} -transients evoked by the stimulus being **not significantly different to the inferred spontaneous spiking activity** (described in detail in the Methods subsection 'Data analysis', beginning: "To define responsive neurons, the spike rates resulting from presentations of orientated grating stimuli and a blank stimulus were subjected to Kruskal-Wallis...").

While the example neuron presented in figure panel 4c met this statistical criteria, it was an error to describe this response as we did. We have consequently modified the description in the text to read (p.11, new text in italics):

"..., which ranged from ineffective (Figure 4a,b) to reducing visual stimulus responses to be not significantly different to spontaneous activity (Figure 4c, Supplementary Figure 9)."

In addition, we have included an additional Supplementary figure (Supplementary Figure 9), referred to in the quote above, in which we present all the individual traces from this example neuron as well as the overlaid average traces.

(Fig. 4e) The panel serves to convey one of the main results, yet it isn't very easy to understand. To facilitate understanding of readers, it is advisable to add another example cell, which shows a significant reduction to the contra stimulation and vc inactivation (open green symbol) in the format similar to panels b and c.

We are happy to add data from the example neuron suggested here by the reviewer, in fact an earlier draft did include this as a panel. However we respectfully feel that adding it into the existing figure makes the figure overly busy, and will make it more confusing. Consequently, we have added the requested additional traces to the manuscript as an additional Supplementary Figure (Supplementary Figure 10) and referred to it in the text as follows (p.11):

*"First we ranked all visually responsive neurons according to the change in their mean firing rate in response to visual stimulation caused by inactivation of the vc pathway (total of 45 neurons significantly modulated, 26 binocular, 5 ipsilateral, and 14 contralateral, Figure 4e, same data as in Figure 4a), and concurrently quantified for each neuron which of the two monocular visual pathways was effected the most (Figure 4e, color coding denotes effect mainly on contra- (more green) or ipsilateral (more red) eye response, *black arrow denotes example in Figure 4c, grey arrow denotes example in Supplementary Figure 10*)."*

What is the advantage of using "delta" as opposed to the "VC reduction index" employed in panel g?

In panel 4e the analysis considered responses to all stimuli (responses to grating at all orientations) as represented by the change in mean firing rate, while in panel g the analysis considers peak responses as represented by the preferred orientation response, used to calculate the vc reduction index.

REVIEWERS' COMMENTS:

Reviewer #1 (Remarks to the Author):

I find the authors' revisions to the manuscript in response to my and the other reviewer's comments to be entirely satisfactory. Nearly all of the comments were addressed to matters of clarity in the presentation, and the only ones to matters of substance have now also been fully address. As I wrote in my original review, the manuscript is fully convincing on its main points, employs adequate methods that have become available only in the past few years, and collects a sufficient body of data to establish the role of inputs from the corpus callosum to the visual cortex in the rat. These are important findings that advance the field considerably. While questions remain about other species, especially the human, I believe that the alternative hypotheses that have been put forward on this question can now be conclusively rejected. The manuscript is suitable for publication in its present form.

Reviewer #2 (Remarks to the Author):

The revised manuscript by Ramachandra and others properly addressed all the queries raised by the reviewer, apart from the query on l.316 of the original manuscript. For this query, the revised manuscript now includes Supplementary Figure 11b, which is addressed in the main text as follows: "...population of monocular neurons in V1. In addition, although callosally projecting axons have been observed to accumulate into patches along the lateral edge of V1 in flattened-cortex histological preparations from rats, we did not observe any clustering of neurons whose responses were more strongly modulated by vc pathway inactivation in an overlay map of all recorded neurons color-coded by their vc reduction index (Supplementary Fig. 11b)."

This figure and associated analysis addressed the query in the requested format. However, the spatial locations of the analyzed neurons in Supplementary Fig11b do not seem to coincide with those of Figure 1i. Also, the number of analyzed neurons does not seem to be equal to that in Figure 1i. Please clarify where the discrepancy stems from. If the analyzed neurons were subsampled, please describe the criterion for it.

Upon a correction of this point, I believe the manuscript is worth publishing in Nature Communications.

Daisuke Shimaoka

Reviewer #2 (Remarks to the Author):

The revised manuscript by Ramachandra and others properly addressed all the queries raised by the reviewer, apart from the query on l.316 of the original manuscript. For this query, the revised manuscript now includes Supplementary Figure 11b, which is addressed in the main text as follows: “...population of monocular neurons in V1. In addition, although callosally projecting axons have been observed to accumulate into patches along the lateral edge of V1 in flattened-cortex histological preparations from rats, we did not observe any clustering of neurons whose responses were more strongly modulated by vc pathway inactivation in an overlay map of all recorded neurons color-coded by their vc reduction index (Supplementary Fig. 11b).”

This figure and associated analysis addressed the query in the requested format. However, the spatial locations of the analyzed neurons in Supplementary Fig11b do not seem to coincide with those of Figure 1i. Also, the number of analyzed neurons does not seem to be equal to that in Figure 1i. Please clarify where the discrepancy stems from. If the analyzed neurons were subsampled, please describe the criterion for it.

Upon a correction of this point, I believe the manuscript is worth publishing in Nature Communications.

Daisuke Shimaoka

We thank Daisuke Shimaoka for this very thorough and helpful review throughout. In response to the comment raised here, the locations are different because in the original Figure 1i the data had been reflected horizontally in the center of the plot to ensure that the cell locations shown in the overlaid panel (Fig. 1i) matched the positions in the overview image shown above it in panel h. This, in turn, results from the different optics used in the intrinsic imaging system compared with the multiphoton microscope. With respect to the number of neurons, the original Fig. 1i showed non-responding neurons as open circles, again for consistency with panel h, while they were omitted from the plot in our revised Supplementary Figure 11. Both were omissions on our part. In the revised version of Supplementary Figure 11b we have transformed the plot so that it is again consistent with the presentation in Figure 1i and have also added in the non-responding cells, again as open circles, for consistency.